# FastCLIP: A Suite of Optimization Techniques to Accelerate CLIP Training with Limited Resources

## Abstract

Existing studies of training state-of-the-art Contrastive Language-Image Pretraining (CLIP) models on large-scale data involve hundreds of or even thousands of GPUs due to the requirement of a large batch size. However, such a large amount of resources is not accessible to most people. While advanced compositional optimization techniques for optimizing global contrastive losses have been demonstrated effective for removing the requirement of a large batch size, their performance on large-scale data remains underexplored and not optimized. To bridge the gap, this paper explores several aspects of CLIP training with **limited resources** (e.g., up to tens of GPUs). First, we introduce FastCLIP, a general CLIP training framework built on advanced compositional optimization techniques while designed and optimized for the distributed setting. Our framework is equipped with an efficient gradient reduction strategy to reduce communication overhead. Second, to further boost training efficiency, we investigate three components of the framework from an optimization perspective: the schedule of the inner learning rate, the update rules of the temperature parameter and the model parameters, respectively. Experiments on different strategies for each component shed light on how to conduct CLIP training more efficiently. Finally, we evaluate the performance of FastCLIP and the state-of-the-art training baseline (OpenCLIP) on different compute scales up to 32 GPUs on 8 nodes, and three data scales ranging from 2.7 million, 9.1 million to 315 million image-text pairs to demonstrate the significant improvement of FastCLIP in the resource-limited setting.

## 1 Introduction

Contrastive Language-Image Pretraining (CLIP) (Radford et al., 2021) is a popular approach for vision-language representation learning (Cherti et al., 2023; Sun et al., 2024; Chen et al., 2023c; Li et al., 2023a; Qiu et al., 2023). The method effectively embeds data from the image and language modality into a joint embedding space by optimizing a contrastive loss in a self-supervised manner. It has demonstrated strong performance on various downstream tasks (e.g., zero-shot classification and retrieval) and has been adopted in various applications, including text-to-image generation (Ramesh et al., 2022; Zhou et al., 2022; Crowson et al., 2022), image captioning (Yu et al., 2022; Mokady et al., 2021), and evaluation of image generation (Hessel et al., 2021). Its popularity is further fueled by releases of web-scale datasets (Schuhmann et al., 2021; 2022; Gadre et al., 2023; Fang et al., 2023).

However, vanilla mini-batch based methods for self-supervised contrastive learning are known to require a large batch size to obtain satisfactory performance (Chen et al., 2023b; 2020). Theoretically, it has been shown that the optimization error of mini-batch based contrastive learning methods inversely depends on the batch size (Yuan et al., 2022). Empirically, state-of-the-art CLIP models are typically trained using a large batch size on a large number of GPUs (e.g., 84k batch size and 1024 Nvidia A100 GPUs in OpenCLIP (Cherti et al., 2023)). Such a large amount of resources is not accessible to most researchers and practitioners in academia and small companies. Recently, Yuan et al. (2022) proposed an algorithm named SogCLR to address the large batch size issue, which leverages **finite-sum coupled compositional optimization (FCCO)** techniques to optimize a global contrastive loss (GCL) that contrasts each anchor data with all other data in a compositional structure. A key feature of compositional optimization is the **inner and outer steps** where the inner steps maintain and update a sequence of estimators to track the inner functions on the solution path, which can be interpreted as an SGD update with a learning rate called the inner learning rate (Wang & Yang,

2022). Later, SogCLR has been leveraged by Qiu et al. (2023) to design the iSogCLR algorithm for optimizing a robust global contrastive loss (RGCL) with individualized learnable temperatures for training CLIP models. However, these algorithms are not fully optimized for large-scale training of CLIP models since they were examined only on small-scale datasets.

This paper aims to scale up the advanced optimization algorithms for optimizing global contrastive losses of CLIP training on large-scale data with limited compute resources. We introduce a distributed training framework named FastCLIP by employing data parallelism such that each worker computes the gradient estimator using their respective data and then reduces (averages) them through communication, based on which the model is updated. A novel gradient reduction strategy is designed, which requires less communication than the existing distributed framework. This distributed training framework lays the foundation for scaling up CLIP training with limited resources. To further boost the efficiency of our framework, we investigate its three aspects from an optimization perspective: the schedule of the inner learning rate (LR) of compositional optimization, the update rule of the temperature parameter, and the update rule of the model parameters, respectively.

- Previous studies (Yuan et al., 2022; Qiu et al., 2023) set the inner LR to a constant value less than but close to one, which could slow down the training for large-scale data at earlier iterations. Inspired by the learning rate schedule of existing optimizers of Deep Learning (Loshchilov & Hutter, 2017), we examine a cosine decay schedule for the inner LR by comparing its performance with the constant schedule.

- For the update rule of the temperature parameter, we compare four different strategies in the FastCLIP framework, including a heuristic approach based on the gradient of GCL, a constant strategy as used in SogCLR, learning individualized temperatures as used in iSogCLR, and learning global temperature by optimizing a new RGCL with a single learnable temperature.

- For the update rule of the model parameters, we compare the performance of commonly-used optimizers for CLIP training in the FastCLIP framework, including AdamW (Loshchilov & Hutter, 2019), LAMB (You et al., 2020), Lion (Chen et al., 2023a) and SGD with momentum (Polyak, 1964).

Moreover, in order to study the scaling capability of FastCLIP, we compare the performance of FastCLIP and state-of-the-art baseline OpenCLIP (Ilharco et al., 2021) on three data scales and four compute scales. The data scales include 2.7 million (CC3M (Sharma et al., 2018)), 9.1 million (CC12M (Changpinyo et al., 2021)), and 315 million (LAION400M (Schuhmann et al., 2021)) image-text pairs[1]. The compute scales include 1, 2, 4, and 8 nodes, with 4 GPUs on each node.

The **contributions** of this paper are summarized as follows: (1) We propose FastCLIP, an efficient distributed framework to scale up CLIP training with limited computing resources. (2) We study the performance of different strategies for three components of FastCLIP, providing insights on how to conduct CLIP training more efficiently. (3) We compare the performance of FastCLIP on different data scales and compute scales. The results show that FastCLIP consistently outperforms state-of-the-art training baseline OpenCLIP by a large margin. A quick comparison between FastCLIP and OpenCLIP on different data scales and compute scales are shown in Figure 1, with more detailed results presented in Section 6.

**Roadmap**: In Section 2 we review the literature of CLIP training, in Section 3 we introduce the objective of interest and provide background on the Global Contrastive Learning framework. We propose our FastCLIP framework and explain its gradient reduction strategy in Section 4. Then in Section 5 we compare different strategies for different components within the FastCLIP framework, and we compare the scaling performance of FastCLIP and OpenCLIP under different settings in Section 6. Finally, we conclude this paper in Sections 7 and 8.

## 2 RELATED WORKS

**CLIP training in the distributed setting**: Radford et al. (2021) train CLIP models in a distributed setting, but few details regarding the implementation are provided. Ilharco et al. (2021) develop OpenCLIP, an open-source implementation of CLIP. They leverage the PyTorch distributed data-parallel module (Li et al., 2020) to automatically communicate features and gradients. EVA-CLIP

---

[1]The size of downloaded sets are smaller than their original versions since some links are no longer valid.

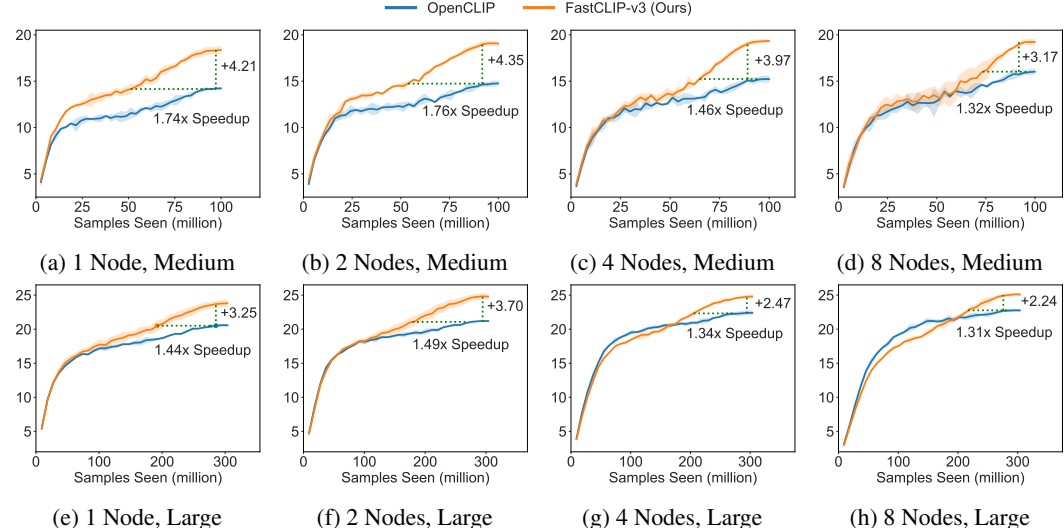

Figure 1: Zero-shot accuracy curves on ImageNet & its variants of OpenCLIP and FastCLIP-v3 trained on 1 to 8 node(s) with 4 GPUs per node on medium and large-scale settings (c.f. Section 5).

(Sun et al., 2023; 2024) scales the number of parameters of the image encoder in CLIP up to 18 billion by applying several techniques from the system perspective, including the ZeRO optimizer (Rajbhandari et al., 2020) and global half-precision training with DeepSpeed (Rasley et al., 2020). The key difference between existing works and this work is that they all use a simple mini-batch based contrastive loss, which suffers from the issue of requiring a large batch size. This in turn requires hundreds and even thousands of GPUs (e.g., 592 V100 in CLIP, 1024 A100 in OpenCLIP, 256 A100 in EVA-CLIP). Our work focuses on scaling up CLIP training in a resource-limited setting with only tens of GPUs.

**Benchmark for CLIP training**: Cherti et al. (2023) study the scaling performance of CLIP training. They measure the performance of CLIP across different model sizes and dataset sizes, and study the relationships between downstream task performance and resource consumption. Gadre et al. (2023) investigate the impact of different data filtering strategies on the trained model's downstream performance. They conduct experiments across different data scales ranging from 12.8 million to 12.8 billion and provide insights on how to curate CLIP's training data. Cui et al. (2022) examine the impact of data quality, supervision strategies (e.g., additional image supervision), and model architectures. Li et al. (2024) explore different aspects of CLIP training under a limited training budget, including the impact of the quality and quantity of the training data, different model architectures, and different existing training strategies. Different from these works, we study different *algorithmic components* of CLIP training in an advanced optimization framework for optimizing the global contrastive loss.

**Improved CLIP training**: Many works have studied efficient CLIP training with limited resources. Yuan et al. (2022) propose SogCLR to improve the performance of contrastive learning with small batch size. Our work scales up SogCLR in the distributed setting and incorporates several algorithmic strategies to accelerate its training speed. Besides the algorithm, other directions are also explored for more efficient CLIP training, including augmenting mini-batch based contrastive losses (Li et al., 2023c; Zhai et al., 2023; Mu et al., 2022; Li et al., 2022; Mo et al., 2023; Lee et al., 2022; Goel et al., 2022), model compression (Wu et al., 2023; Li et al., 2023b; Fang et al., 2021), and system optimization (Chen et al., 2023b; Sun et al., 2023; Rajbhandari et al., 2020).

**Temperature scheme**: The temperature parameter in contrastive losses plays an important role in CLIP training. Many techniques have been proposed to update or set the temperature parameter. Radford et al. (2021) treat the temperature as part of the learnable parameters in the mini-batch contrastive loss. Zhang et al. (2022) propose to use different temperatures for positive and negative samples to independently control intra-anchor and inter-anchor hardness-awareness. Kukleva et al. (2023) study a cosine decay schedule for setting the temperature. Huang et al. (2023b) propose to set the temperature parameter proportional to the alignment between positive pairs. Qiu et al. (2023) propose a robust global contrastive loss (RGCL) with individualized temperatures inspired by

Distributionally Robust Optimization and optimize it with the iSogCLR algorithm which extends SogCLR. However, their performance on large-scale data remains unknown. This work discovers a new strategy by learning a global temperature in the RGCL that yields better performance for large-scale data.

**Optimizers for CLIP training**: Different optimizers for updating the learnable parameters have been employed in CLIP training, including AdamW (Loshchilov & Hutter, 2019) used by Radford et al. (2021); Cherti et al. (2023); Gadre et al. (2023); Chen et al. (2023c); Li et al. (2023a); Qiu et al. (2023), and LAMB (You et al., 2020) used by Sun et al. (2023); Xie et al. (2023); Huang et al. (2023a). A recently proposed optimizer named Lion (Chen et al., 2023a) also provides promising results for this task (Chen et al., 2023a; Wortsman et al., 2023). In this work, we compare the performance of AdamW, LAMB and Lion to determine which optimizer is most suitable in FastCLIP for training CLIP models from scratch. We also include SGD with momentum (Polyak, 1964) for comparison.

## 3 PRELIMINARIES

**Notations**: Given a dataset $\mathcal{S}$ of $n$ images $\boldsymbol{x}_i$ and their corresponding text descriptions $\boldsymbol{z}_i$: $\mathcal{S} = \{(\boldsymbol{x}_1, \boldsymbol{z}_1), \ldots, (\boldsymbol{x}_n, \boldsymbol{z}_n)\}$, we aim to learn an image encoder and a text encoder (jointly represented by $\boldsymbol{w}$) from the data. We use $\boldsymbol{e}_{1,i} = \boldsymbol{e}_1(\boldsymbol{w}, \boldsymbol{x}_i) \in \mathbb{R}^d$ and $\boldsymbol{e}_{2,i} = \boldsymbol{e}_2(\boldsymbol{w}, \boldsymbol{z}_i) \in \mathbb{R}^d$ to denote the encoded vector of the input $\boldsymbol{x}_i$ and $\boldsymbol{z}_i$, respectively. And we use $\boldsymbol{e}_i = (\boldsymbol{e}_{1,i}^\top, \boldsymbol{e}_{2,i}^\top)^\top$ to denote the concatenation of $\boldsymbol{e}_{1,i}$ and $\boldsymbol{e}_{2,i}$. Denote by $\mathcal{B} \subset \mathcal{S}$ a mini-batch of image-text pairs. With slight abuse of notation, we also use $\mathcal{B}$ (and $\mathcal{S}$) to denote the indices of the image-text pairs it contains. $\mathcal{S}_{i-} := \mathcal{S} \backslash \{i\}$ denotes the subset of $\mathcal{S}$ without $i$-th pair. We consider the data parallel setting such that $\mathcal{S}$ is partitioned evenly across $K$ workers denoted by $\mathcal{S}_1, \ldots, \mathcal{S}_K$. For a function $\ell(\cdot, \cdot)$, let $\nabla_1 \ell(\cdot, \cdot)$ and $\nabla_2 \ell(\cdot, \cdot)$ denote the partial gradient in terms of the first and second argument, respectively.

**Mini-batch Contrastive Loss (MBCL) and Global Contrastive Loss (GCL)**: The core idea of CLIP training is to leverage a contrastive loss to push features of paired image and text close to each other (i.e., to maximize the similarity between $\boldsymbol{e}_{1,i}$ and $\boldsymbol{e}_{2,i}$), while pushing features of non-paired image and text away from each other (i.e., minimizing the similarity between $\boldsymbol{e}_{1,i}$ and $\boldsymbol{e}_{2,j}$ for $i \neq j$). Mathematically, let $s_{i,j}$ denote the cosine similarity between $\boldsymbol{e}_{1,i}$ and $\boldsymbol{e}_{2,j}$. Define

$$\ell_1(\boldsymbol{e}_i, \boldsymbol{e}_{2,j}, \tau) := \exp\left((s_{i,j} - s_{i,i})/\tau\right), \quad \ell_2(\boldsymbol{e}_i, \boldsymbol{e}_{1,j}, \tau) := \exp\left((s_{j,i} - s_{i,i})/\tau\right),$$

where $\tau > 0$ is the temperature parameter. Given a mini-batch $\mathcal{B}$ of image-text pairs, let

$$g_1(\boldsymbol{w}, \tau, i, \mathcal{B}) := \frac{1}{|\mathcal{B}|} \sum_{j \in \mathcal{B}} \ell_1(\boldsymbol{e}_i, \boldsymbol{e}_{2,j}, \tau), \quad g_2(\boldsymbol{w}, \tau, i, \mathcal{B}) := \frac{1}{|\mathcal{B}|} \sum_{j \in \mathcal{B}} \ell_2(\boldsymbol{e}_i, \boldsymbol{e}_{1,j}, \tau).$$

In the literature, a large number of works (e.g., Cherti et al., 2023; Gadre et al., 2023; Sun et al., 2023), following Radford et al. (2021), minimize the mini-batch contrastive loss (MBCL):

$$\frac{1}{|\mathcal{S}|} \sum_{i \in \mathcal{S}} \mathbb{E}_{\mathcal{B} \subset \mathcal{S}_{i-}} \left( \log\left(\frac{1}{|\mathcal{B}|} + g_1(\boldsymbol{w}, \tau, i, \mathcal{B})\right) + \log\left(\frac{1}{|\mathcal{B}|} + g_2(\boldsymbol{w}, \tau, i, \mathcal{B})\right) \right), \quad \text{(MBCL)}$$

which contrasts the $i$-th pair with other pairs within only a mini-batch $\mathcal{B}$. However, this loss suffers from the large-batch size issue, which has been addressed by the Global Contrastive Loss (GCL) (Yuan et al., 2022) that contrasts the $i$-th pair with all other pairs in the dataset $\mathcal{S}$:

$$\frac{\tau}{|\mathcal{S}|} \sum_{i \in \mathcal{S}} \left( \log\left(\varepsilon + g_1(\boldsymbol{w}, \tau, i, \mathcal{S}_{i-})\right) + \log\left(\varepsilon + g_2(\boldsymbol{w}, \tau, i, \mathcal{S}_{i-})\right) \right), \quad \text{(GCL)}$$

where $\varepsilon$ is a small constant.

**Robust Global Contrastive Loss (RGCL)**: To improve CLIP training, Qiu et al. (2023) designed a robust global contrastive loss (RGCL) with individualized temperature parameters inspired by Distributionally Robust Optimization. It is defined as:

$$\min_{\tau_1, \tau_2 \geq \tau_0} \frac{1}{|\mathcal{S}|} \sum_{i \in \mathcal{S}} \left( \tau_{1,i} \cdot \left(\log\left(\varepsilon + g_1(\boldsymbol{w}, \tau_{1,i}, i, \mathcal{S}_{i-})\right) + \rho\right) \right.$$
$$\left. + \tau_{2,i} \cdot \left(\log\left(\varepsilon + g_2(\boldsymbol{w}, \tau_{2,i}, i, \mathcal{S}_{i-})\right) + \rho\right) \right), \quad \text{(RGCL)}$$

where $\tau_1 = (\tau_{1,1}, \ldots, \tau_{1,n})$, $\tau_2 = (\tau_{2,1}, \ldots, \tau_{2,n})$, $\tau_0$ is a small value, $\rho \geq 0$ is a hyperparameter.

**Optimization Algorithms.** To optimize GCL, Yuan et al. (2022) proposed the SogCLR algorithm based on advanced compositional optimization known as Finite-sum Coupled Compositional Optimization (FCCO) (Wang & Yang, 2022). Specifically, GCL is formulated as $\frac{1}{n} \sum_{i \in \mathcal{S}} f(g_i(\boldsymbol{w}))$,

---

**Algorithm 1:** The FastCLIP Framework (Sketch)

1 **Input:** Initial model parameters $\boldsymbol{w}^0, \tau^0, (\boldsymbol{u}_1^0, \boldsymbol{u}_2^0)$, Number of iterations $T$.
2 **for** $t = 0, \ldots, T-1$ **do**
3     **for** *each worker* $k$ **do in parallel**
4         Sample a batch $\mathcal{B}_k^t$ from $\mathcal{S}_k$ and compute features $\mathcal{E}_k^t = \{(\boldsymbol{e}_{1,j}, \boldsymbol{e}_{2,j})\}_{j \in \mathcal{B}_k^t}$
5         ALL_GATHER $\mathcal{E}^t = \cup_k \mathcal{E}_k^t$ to obtain global features
6         Compute mini-batch contrastive losses $g_{1,i}^t, g_{2,i}^t$ for $i \in \mathcal{B}_k^t$ (c.f. Proc. 2 in Appendix A)
7         Update $u_{1,i}^{t+1}, u_{2,i}^{t+1}$ using Eqn. (1) for $i \in \mathcal{B}_k^t$. Set $u_{1,i}^{t+1} = u_{1,i}^t, u_{2,i}^{t+1} = u_{2,i}^t$ for $i \notin \mathcal{B}_k^t$
8         Set $\mathcal{U}_k^t = \{(u_{1,j}^{t+1}, u_{2,j}^{t+1})\}_{j \in \mathcal{B}_k^t}$, and ALL_GATHER $\mathcal{U}^t = \cup_k \mathcal{U}_k^t$
9         Compute gradient estimators $G_{\boldsymbol{w},k}^t$ for $\boldsymbol{w}$ using techniques of FCCO (c.f. Proc. 3)
10        ALL_REDUCE $G_{\boldsymbol{w}}^t = \frac{1}{K} \sum_{l=1}^K G_{\boldsymbol{w},l}^t$ across all workers
11        Update $\boldsymbol{w}^{t+1}$ from $\boldsymbol{w}^t$ using an optimizer (c.f. Proc. 4).
12        Update $\tau^{t+1}$ from $\tau^t$ (c.f. Proc. 5).

---

where $f(g) = \log(\varepsilon + g)$ and $g_i(\boldsymbol{w})$ is the inner function inside the log. The main challenge is to compute a gradient estimator using a mini-batch of samples such that the algorithm can converge without requiring a large batch size. The key idea of SogCLR is to maintain and update an estimator for each inner function $g_i(\boldsymbol{w})$ denoted by $u_i$, by using Equation (1). As a result, the gradient at the $t$-th iteration is estimated by $\frac{1}{|\mathcal{B}|} \sum_{i \in \mathcal{B}} \nabla f(u_i^{t+1}) \nabla \hat{g}_i(\boldsymbol{w}^t)$, where $\mathcal{B}$ is a mini-batch and $\hat{g}_i(\boldsymbol{w})$ is a mini-batch estimator of $g_i(\boldsymbol{w})$. To optimize RGCL, Qiu et al. (2023) proposed the iSogCLR algorithm by combining SogCLR with stochastic coordinate updates for the temperature parameters.

## 4 FASTCLIP: A DISTRIBUTED TRAINING FRAMEWORK OF CLIP MODELS

FastCLIP is a distributed training framework for optimizing global contrastive losses (including (GCL) and (RGCL)). Its key updates are built upon the SogCLR algorithm. The main difference between SogCLR and mini-batch based methods such as CLIP is that SogCLR maintains two scalar sequences $u_{1,i}$ and $u_{2,i}$ to keep track of $g_1(\boldsymbol{w}, \tau, i, \mathcal{S}_{i-})$ and $g_2(\boldsymbol{w}, \tau, i, \mathcal{S}_{i-})$ as stated in Section 3. At iteration $t$, for $i$ selected in the batch $\mathcal{B}^t$, $u_{1,i}$ and $u_{2,i}$ will be updated using a moving average estimator with hyperparameter $\gamma_t \in (0, 1]$:

$$u_{1,i}^{t+1} = (1 - \gamma_t)u_{1,i}^t + \gamma_t g_1(\boldsymbol{w}^t, \tau^t, i, \mathcal{B}_{i-}^t), \quad u_{2,i}^{t+1} = (1 - \gamma_t)u_{2,i}^t + \gamma_t g_2(\boldsymbol{w}^t, \tau^t, i, \mathcal{B}_{i-}^t), \quad (1)$$

and the gradient estimator is computed by $\frac{1}{|\mathcal{B}^t|} \sum_{i \in \mathcal{B}^t} \nabla f(u_i^{t+1}) \nabla \hat{g}_i(\boldsymbol{w}^t)$. The core of FastCLIP (Algorithm 1) is how to compute the gradient estimator in a distributed manner.

Next, we use (GCL) as an example to present our gradient computation strategy that effectively reduces the communication cost. We only present key steps and defer the complete derivation to Appendix A due to space limit. Let $\mathcal{B}_k^t$ denote local mini-batch on $k$-th worker. Below, we omit the superscript $t$ and use $\mathcal{B}_k$ for simplicity. Note that (GCL) is the sum of two parts: the image part (loss $g_1$) and the text part (loss $g_2$). Due to their symmetric structure, we only present the gradient of the image part. The gradient estimator of (GCL) is computed by $G_{\boldsymbol{w},1,a} + G_{\boldsymbol{w},1,b}$:

$$G_{\boldsymbol{w},1,a} = \tau \cdot \underbrace{\frac{1}{K}\sum_{k=1}^K}_{\text{ALL\_REDUCE}} \frac{1}{|\mathcal{B}_k|}\sum_{i \in \mathcal{B}_k} \underbrace{\frac{1}{\varepsilon + u_{1,i}}}_{\text{local}} \cdot \overbrace{\frac{1}{K}\sum_{k'=1}^K \frac{1}{|\mathcal{B}_{k',i-}|}\sum_{j \in \mathcal{B}_{k',i-}} \nabla_1 \ell_1(\underbrace{\boldsymbol{e}_i}_{\text{local}}, \underbrace{\boldsymbol{e}_{2,j}}_{\text{global}}, \tau) \cdot \underbrace{\nabla \boldsymbol{e}_i}_{\text{local}}}^{G_{\boldsymbol{w},1,a,i}},$$

$$G_{\boldsymbol{w},1,b} = \tau \cdot \underbrace{\frac{1}{K}\sum_{k'=1}^K}_{\text{ALL\_REDUCE}} \frac{1}{|\mathcal{B}_{k'}|}\sum_{j \in \mathcal{B}_{k'}} \cdot \frac{1}{K}\sum_{k=1}^K \frac{1}{|\mathcal{B}_{k,j-}|}\sum_{i \in \mathcal{B}_{k,j-}} \underbrace{\frac{1}{\varepsilon + u_{1,i}}}_{\text{global}}\nabla_2 \ell_1(\underbrace{\boldsymbol{e}_i}_{\text{global}}, \underbrace{\boldsymbol{e}_{2,j}}_{\text{local}}, \tau) \cdot \underbrace{\nabla \boldsymbol{e}_{2,j}}_{\text{local}}.$$

Both $G_{\boldsymbol{w},1,a}$ and $G_{\boldsymbol{w},1,b}$ have two averages over $\mathcal{B}$ due to compositional structure of the loss. For FastCLIP, the inner average (e.g. $G_{\boldsymbol{w},1,a,i}$) is computed on a single worker after gathering global parts (shaded, e.g., $\boldsymbol{e}_{2,j}$) from all workers. The outer average is then computed using ALL_REDUCE.

Table 1: Comparison between different algorithms. In Temperature Scheme, "G" denotes global temperature parameter, while "I" denotes individualized temperature parameters for each data.

| Algorithm | Loss | FCCO | Distributed | Inner LR Schedule | Temperature Scheme |
|---|---|---|---|---|---|
| OpenCLIP (Ilharco et al., 2021) | (MBCL) | No | Yes | N/A | G, Learnable |
| SogCLR (Yuan et al., 2022) | (GCL) | Yes | No | Constant | G, Constant |
| iSogCLR (Qiu et al., 2023) | (RGCL) | Yes | No | Constant | I, Learnable |
| FastCLIP-v0 | (GCL) | Yes | Yes | Cosine | G, Learnable |
| FastCLIP-v1 | (GCL) | Yes | Yes | Cosine | G, Constant |
| FastCLIP-v2 | (RGCL) | Yes | Yes | Cosine | I, Learnable |
| FastCLIP-v3 | (RGCL-g) | Yes | Yes | Cosine | G, Learnable |

**Difference from OpenCLIP.** Algorithmically, OpenCLIP does not use the $u$ sequence, which is equivalent to setting $\gamma_t = 1$. In terms of distributed implementation, for computing $G_{\boldsymbol{w},1,b}$, OpenCLIP first computes $\frac{1}{\varepsilon + u_{1,i}} \nabla_2 \ell_1(\boldsymbol{e}_i, \boldsymbol{e}_{2,j}, \tau)$ on the worker where $i$-th pair resides, then all workers gather them using REDUCE_SCATTER and uses them to compute the inner average.

FastCLIP has the same communication and computation cost for computing $G_{\boldsymbol{w},1,a}$ as OpenCLIP, but has an effective communication reduction for computing $G_{\boldsymbol{w},1,b}$. Specifically, REDUCE_SCATTER in OpenCLIP requires $\mathcal{O}(K|\mathcal{B}|d)$ communication cost, where $d$ is the feature dimensionality (>512 in practice). While ALL_GATHER of $u_{1,i}$ in FastCLIP requires only $\mathcal{O}(K|\mathcal{B}|)$ communication since each $u_{1,i}$ is a scalar. This leads to a communication reduction, as verified empirically in Sec. 6.

## 5 IMPROVEMENT OF OPTIMIZATION COMPONENTS

In this section, we propose different strategies to improve three components of the FastCLIP framework, i.e., the schedule for inner LR $\gamma_t$, the update rule of the temperature parameter, and the optimizer for updating the model parameters.

**The Inner LR Schedule**: We first explore different schedules for $\gamma_t$ in Equation (1), which is interpreted as an SGD step with learning rate (LR) $\gamma_t$ by Wang & Yang (2022). They showed in theory that $\gamma_t$ should be set to a very small value close to 0 in order to guarantee convergence. However, in practice a large $\gamma_t$ value close to 1 is adopted (Yuan et al., 2022). Ideally, $\gamma_t$ should be large to rely more on the current mini-batch at earlier iterations and be smaller to rely more on history in later iterations. To achieve this, we consider a cosine schedule to decrease $\gamma_t$: Let $t$ be the current iteration, $\hat{E}$ be the number of iterations per epoch and $E$ be the number of decay epochs, then we set $\gamma_t = 0.5 \cdot (1 + \cos(\pi \lfloor t/\hat{E} \rfloor / E)) \cdot (1 - \gamma_{\min}) + \gamma_{\min}$. With this schedule, $\gamma_t$ will decrease from 1.0 to $\gamma_{\min}$. Note that $\lfloor t/\hat{E} \rfloor$ denotes the current epoch, which means the value of $\gamma_t$ stays unchanged within one epoch. Also, The number of decay epochs $E$ is a hyperparameter, and it is not necessarily equal to the total number of training epochs. If the current epoch exceeds $E$, $\gamma_t$ will be set to $\gamma_{\min}$.

**The Temperature Parameter Updates**: At Line 12 of Algorithm 1, the temperature parameter $\tau$ is updated. The update rule is not explicitly provided due to its variety. We consider four different versions, named v0 to v3. Specifically, v1 sets $\tau$ to a constant as in SogCLR and the other three view $\tau$ as a learnable parameter: v2 leverages the same $\tau$ update as iSogCLR, which maintains individual temperature parameters for each data and updates them using gradient of (RGCL) w.r.t. $\tau$. A potential issue of maintaining and updating individualized temperature is that it may overfit the data and hence harm the generalization for large-scale data. To mitigate this issue, we also consider the following loss, which unifies the individual temperature in (RGCL) into a single global one:

$$\min_{\tau \geq \tau_0} \frac{\tau}{|\mathcal{S}|} \sum_{i \in \mathcal{S}} \left( \log\left(\varepsilon + g_1(\boldsymbol{w}, \tau, i, \mathcal{S}_{i-})\right) + \log\left(\varepsilon + g_2(\boldsymbol{w}, \tau, i, \mathcal{S}_{i-})\right) \right) + 2\rho\tau. \quad \text{(RGCL-g)}$$

We refer to this version as v3. We also include a baseline version named v0 that updates $\tau$ using the gradient of an unscaled version of (GCL) that does not multiply $\tau$, similar to the $\tau$ updates in existing works (Radford et al., 2021; Cherti et al., 2023) based on (MBCL). The explicit rules of all updates are deferred to Proc. 5 in Appendix A. Combining the four versions of updating/setting $\tau$ with the cosine inner LR schedule, we get four algorithms FastCLIP-v0 to v3. A comparison between them and existing algorithms is shown in Table 1. Different updates of $\tau$ also lead to slightly different ways of computing the contrastive losses and gradient estimator (Line 6 and Line 9 in Algorithm 1), and the details are deferred to Appendix A due to space limit.

Table 2: Overview of the experiment settings. # Samples denotes the size of the dataset downloaded. Batch Size denotes per-GPU batch size, with global batch size specified in parentheses. H100 has 80GB memory. Some epochs of tested algorithms for the xlarge-scale setting were run on a different system with 16xA100 (40GB) using a local batch size 320. We rename the downloaded 315M subset of LAION400M (Schuhmann et al., 2021) as LAION315M to indicate its actual size.

| Setting | Dataset | # Samples/Epochs | Vision Encoder | Batch Size | GPUs |
|---------|---------|------------------|----------------|------------|------|
| Medium | CC3M | 2.7M/37 epochs | ResNet50 | 128 (1024) | 8 Tesla T4 |
| Large | CC12M | 9.1M/33 epochs | ViT-B/32 | 256 (2048) | 8 Tesla T4 |
| xLarge | LAION315M | 315M/42 epochs | ViT-B/16 | 640 (5120) | 8 H100 |

**The Optimizer**: We compare the performance of four optmizers (i.e., the update rule of model parameters and temperature at Line 11 to 12 in Algorithm 1) in FastCLIP: AdamW (Loshchilov & Hutter, 2019), LAMB (You et al., 2020), Lion (Chen et al., 2023a) and SGD with momentum (Polyak, 1964). The update rules of these optimizers are presented in Proc. 4 in Appendix A for completeness.

**Experiment Settings**: We conduct experiments in three different settings, which differ in data scale, model architecture (vision encoder in particular), and training environment. The difference is presented in Table 2. In all settings, we use a 12-layer transformer (Vaswani et al., 2017) as the text encoder. All the experiments are conducted in a multi-node setting where each node has 4 GPUs. Due to its extreme size, xlarge-scale setting is only used to compare the best version of FastCLIP with OpenCLIP. The value of $\varepsilon$ is set to 1e-14 for all but the xlarge-scale setting, where we use a large value of 1e-6. This is discussed in Section 6 and Appendix D.

**Metrics**: To evaluate the performance of the trained models, we leverage the Datacomp Benchmark (Gadre et al., 2023), which includes 38 zero-shot downstream tasks. The evaluation metric is the average performance, which is called Datacomp. We also report the average performance on two subsets of the tasks: ImageNet and its different variants (IN & Variants), and Retrieval. IN & Variants consists of top 1 accuracy on ImageNet-1k (Deng et al., 2009) and 6 ImageNet distribution shift datasets (Wang et al., 2019; Recht et al., 2019; Hendrycks et al., 2021b;a; Barbu et al., 2019) (Gadre et al., 2023, Section 3.5). Retrieval consists of mean recall at 1 on Flickr30k (Young et al., 2014), MSCOCO (Chen et al., 2015), and jaccard score on WinoGAViL (Bitton et al., 2022). We refer the readers to Gadre et al. (2023) for detail of all the tasks.

## 5.1 RESULTS

In this subsection, we present the experiment results. We report results averaged over 3 runs with different seeds, and standard deviation in parentheses. Training details are provided in Appendix B.

**The Inner LR Schedule**: We first present results of different $\gamma$ schedules. We compare three pairs of approaches: SogCLR and FastCLIP-v1; iSogCLR and FastCLIP-v2; FastCLIP-v3 with Constant $\gamma$ and FastCLIP-v3, where the former of each pair uses constant $\gamma$ schedule and the latter uses cosine $\gamma$ schedule. SogCLR and iSogCLR are implemented in the same framework as FastCLIP. The results are presented in Table 3. We can observe that all of the three approaches obtain a significant performance gain when equipped with the cosine schedule. This indicates that cosine schedule performs better than the constant schedule. Also, when tuning the $\gamma$ value for the two schedules, we observe that constant schedule favors larger $\gamma$ values (0.6 or 0.8), while cosine schedule favors small $\gamma$ value (0.2) in the end (c.f. Table 8 in Appendix B). These results suggest: (1) $\gamma$ needs to be set to a small value as the theory predicts, (2) but instead of being constant, its value should decrease.

**The Temperature Parameter Updates**: Next, we present the experiment results of different $\tau$ updates. We compare the four versions of FastCLIP. The results are presented in Table 4. We have the following observations. In the medium-scale setting, the average performance on Datacomp of the four algorithms are close to each other. FastCLIP-v3 has better performance than others either on Retrieval or IN & Variants. In the large-scale setting, FastCLIP-v3 outperforms other algorithms on Datacomp and Retrieval. This demonstrates the effectiveness of FastCLIP-v3. Also we can see that FastCLIP-v0, v2 are comparable to each other while FastCLIP-v1 is generally worse in this setting.

**The Optimizer**: We use FastCLIP-v3 as the base algorithm and compare the AdamW, LAMB, Lion and SGD with momentum optimizers. The results are presented in Table 5. We observe that AdamW

Table 3: Performance of different inner LR schedules. Shaded algorithms use the cosine schedule, while the others use the constant schedule. Improvement denotes the absolute difference between two algorithms on the three metrics. *: v3 (Const. $\gamma$) denotes FastCLIP-v3 with constant $\gamma$ schedule. The meaning of each metric is provided in Section 5.

| Setting | Algorithm | Datacomp | Retrieval | IN & Variants | *Improvement* |
|---------|-----------|----------|-----------|---------------|---------------|
| Medium | SogCLR | 23.41 (0.34) | 27.48 (0.24) | 16.90 (0.01) | *1.46, 1.80, 1.96* |
| | FastCLIP-v1 | **24.87** (0.13) | **29.28** (0.30) | **18.86** (0.09) | |
| | iSogCLR | 23.35 (0.63) | 27.92 (0.34) | 17.05 (0.14) | *0.75, 1.40, 1.47* |
| | FastCLIP-v2 | **24.10** (0.34) | **29.32** (1.29) | **18.52** (0.37) | |
| | v3 (Const. $\gamma$)* | 23.60 (0.18) | 27.68 (0.17) | 17.33 (0.22) | *1.16, 2.68, 1.75* |
| | FastCLIP-v3 | **24.76** (0.26) | **30.36** (0.18) | **19.08** (0.16) | |
| Large | SogCLR | 29.91 (0.23) | 30.16 (0.36) | 22.98 (0.07) | *0.74, 2.50, 1.28* |
| | FastCLIP-v1 | **30.65** (0.11) | **32.66** (0.12) | **24.26** (0.06) | |
| | iSogCLR | 30.32 (0.18) | 30.27 (0.41) | 24.96 (0.09) | *0.62, 1.57, 0.56* |
| | FastCLIP-v2 | **30.94** (0.20) | **31.84** (0.17) | **25.52** (0.17) | |
| | v3 (Const. $\gamma$)* | 29.46 (0.39) | 30.33 (0.58) | 23.69 (0.09) | *2.14, 4.55, 1.09* |
| | FastCLIP-v3 | **31.60** (0.46) | **34.88** (0.28) | **24.78** (0.28) | |

Table 4: Performance of different temperature parameter updates. All algorithms use AdamW as the optimizer. The meaning of each metric is provided in Section 5.

| Setting | Algorithm | Datacomp | Retrieval | IN & Variants |
|---------|-----------|----------|-----------|---------------|
| Medium | FastCLIP-v0 | 24.71 (0.21) | 30.36 (0.26) | 17.50 (0.33) |
| | FastCLIP-v1 | **24.87 (0.13)** | 29.28 (0.30) | 18.86 (0.09) |
| | FastCLIP-v2 | 24.21 (0.76) | 30.35 (0.47) | 17.86 (0.21) |
| | FastCLIP-v3 | 24.76 (0.26) | **30.36 (0.18)** | **19.08 (0.16)** |
| Large | FastCLIP-v0 | 31.47 (0.31) | 34.86 (0.53) | 24.55 (0.21) |
| | FastCLIP-v1 | 30.65 (0.11) | 32.66 (0.12) | 24.26 (0.06) |
| | FastCLIP-v2 | 30.95 (0.32) | 33.71 (0.20) | **24.94 (0.18)** |
| | FastCLIP-v3 | **31.60 (0.46)** | **34.88 (0.28)** | 24.78 (0.28) |

Table 5: Performance of different optimizers. SGDM denotes SGD with momentum. The base algorithm is FastCLIP-v3 for all optimizers. The meaning of each metric is provided in Section 5.

| Setting | Algorithm | Datacomp | Retrieval | IN & Variants |
|---------|-----------|----------|-----------|---------------|
| Medium | SGDM | 22.25 (0.13) | 26.06 (0.03) | 16.32 (0.06) |
| | LAMB | 22.63 (0.30) | 24.87 (0.27) | 16.43 (0.06) |
| | Lion | 24.50 (0.12) | 29.41 (0.26) | 18.03 (0.10) |
| | AdamW | **24.76** (0.26) | **30.36** (0.18) | **19.08** (0.16) |
| Large | SGDM | 30.15 (0.48) | 33.09 (0.28) | 22.95 (0.22) |
| | LAMB | 30.54 (0.24) | 34.02 (0.26) | 24.11 (0.21) |
| | Lion | 30.99 (0.09) | 33.78 (0.22) | **25.01** (0.18) |
| | AdamW | **31.60** (0.46) | **34.88** (0.28) | 24.78 (0.28) |

outperforms other optimizers on most of the metrics in both settings. This indicates that AdamW should be chosen for FastCLIP training.

# 6 SCALING PERFORMANCE OF FASTCLIP

In this section, we compare the performance of FastCLIP using AdamW on different number of nodes in comparison with OpenCLIP. We conduct experiments on 1, 2, 4, and 8 node(s). Except for the number of nodes, other settings are kept the same as the experiment settings specified in Section 5. Training details and additional experiment results are provided in Appendix B and E, respectively.

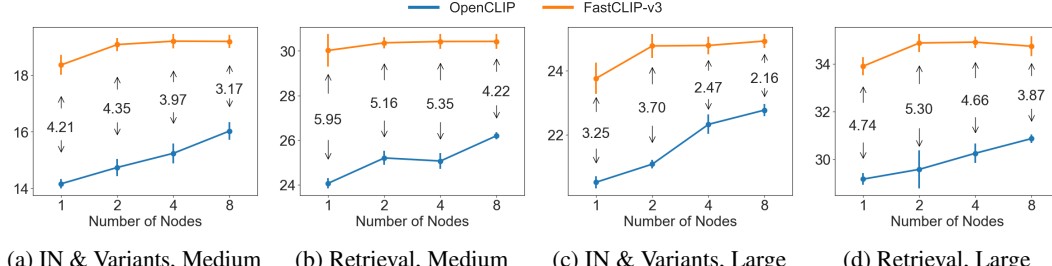

(a) IN & Variants, Medium    (b) Retrieval, Medium    (c) IN & Variants, Large    (d) Retrieval, Large

Figure 2: Comparison between OpenCLIP and FastCLIP-v3. The numbers in between represent the improvement of FastCLIP-v3 over OpenCLIP.

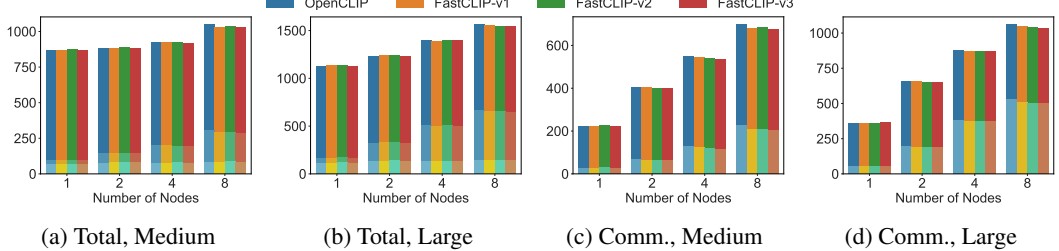

(a) Total, Medium    (b) Total, Large    (c) Comm., Medium    (d) Comm., Large

Figure 3: Comparison of per-iteration running time (ms) between OpenCLIP and FastCLIP. Each bar in (a), (b) is divided into three parts (top to bottom): computation, communication (not overlapped with computation), and others. Each bar in (c), (d) is divided into two pars (top to bottom): communication-computation overlap and pure communication.

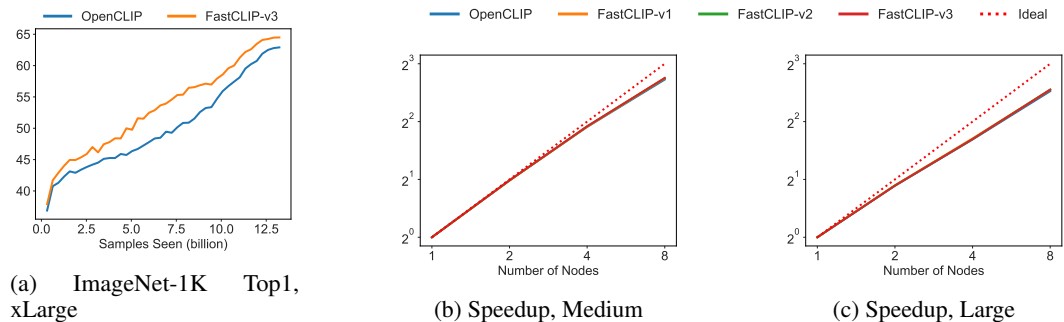

(a)    ImageNet-1K    Top1, xLarge    (b) Speedup, Medium    (c) Speedup, Large

Figure 4: Subfigure (a) presents the ImageNet-1k Top1 accuracy curve of OpenCLIP and FastCLIP-v3 in the xLarge-scale setting, with numbers denoting the improvement. Subfigures (b), (c) present the speedup of different algorithms in the medium and large-scale settings, respectively.

**Performance**: The results of selected models based on the average Datacomp performance are presented in Figure 2, Subfigures (a) and (b) are the IN & Variants and Retrieval performance in the medium-scale setting, and subfigures (c) and (d) are the results in the large-scale setting. We can observe that FastCLIP-v3 consistently outperforms OpenCLIP across different number of nodes. This clearly illustrates the advantage of GCL family over MBCL. Also, the performance of FastCLIP-v3 plateaus at 2 nodes, which verifies that FastCLIP does not require a large amount of computing resources. In contrast, OpenCLIP has a significant performance gain when scaling from 2 nodes to 8 nodes, meaning that it requires a large amount of computing resources to obtain good performance. Additionally, Figure 1 demonstrates the significant speedup of FastCLIP-v3 over OpenCLIP.

**Training Time**: In addition to the performance on downstream tasks, we also compare the training time of OpenCLIP and FastCLIP-v1 to v3. We use PyTorch (Paszke et al., 2019) Profiler to record the data. We break down per-iteration training time into 3 parts: computation, pure communication (not overlapped with computation), and others. The results are plotted in Figure 3 (a) and (b). We also break down communication into two parts: communication overlapped with computation and pure communication, which are plotted in Figure 3 (c) and (d). From subfigures (a) and (b) we can see that the running time of FastCLIP is similar to OpenCLIP when the number of nodes is small (1

and 2), and becomes shorter than OpenCLIP when the number of nodes scales up (4 and 8). This is because OpenCLIP has a longer communication time on 4 and 8 nodes (subfigures (c) and (d)), which demonstrates the effectiveness of our efficient gradient computation/communication strategy described in Section 4. The above results are obtained from a cluster with InfiniBand interconnect. We also profile the training time of the algorithms on two other clusters with Slingshot interconnect, where we observe the same trend. We defer the additional results to Appendix E due to space limit. For each algorithm, we also plot its speedup over 1 node in terms of training time in Figure 4 (a) and (b). All algorithms have similar speedup over 1 node and the gap between the ideal speedup (which is number of nodes) and the real speedup becomes larger when the number of nodes scales up. This indicates that training with more resources has a diminishing return.

Table 6: Summary of existing and our results of training CLIP models on xlarge-scale data.

| Work | Architecture | Data Size (M) | Batch Size | Samples (B) | IN 0-shot (%) |
|------|-------------|---------------|------------|-------------|---------------|
| Cherti et al. (2023) | ViT-B/16 | 80 | 90112 | 13 | 60.24 |
| Cherti et al. (2023) | ViT-B/16 | 400 | 33792 | 13 | 67.00 |
| Cherti et al. (2023) | ViT-B/16 | 2000 | 90112 | 13 | 68.13 |
| Chen et al. (2023b) | ViT-B/32 | 400 | 65536 | 13 | 64.30 |
| OpenCLIP (our impl.) | ViT-B/16 | 315 | 5120 | 13 | 62.90 |
| FastCLIP-v3 | ViT-B/16 | 315 | 5120 | 13 | 64.49 |

**Results in the xlarge-scale setting.** Moreover, we evaluate the performance of FastCLIP-v3 and OpenCLIP in the xlarge-scale setting with 8 H100 GPUs. We plot the ImageNet-1k top 1 accuracy curve in Figure 4 (a). After seeing 13B examples, OpenCLIP achieves a top1 accuracy of 62.90% on ImageNet-1k, while FastCLIP-v3 achieves an accuracy of 64.49%, resulting in a 1.59% gain. This result is competitive with the state-of-the-art results of CLIP training using much more compute resources as shown in Table 6. We also note that the result of our OpenCLIP implementation is lower than those reported in other works, e.g., 67% in OpenCLIP paper that uses a batch size of 33,792 and 400M dataset (Cherti et al., 2023). This is because in our setting we use a smaller dataset (315M) and a smaller batch size (5120). We provide a discussion of the impact of dataset size and batch size in Appendix C. For FastCLIP-v3 in the xlarge-scale setting, we found that assigning a larger value of 1e-6 to the constant $\varepsilon$ than the default 1e-14 in loss computation of (RGCL-g) leads to improved ImageNet-1k top 1 accuracy. We provide a brief discussion of this observation in Appendix D. We also evaluate the Datacomp performance of FastCLIP-v3 and OpenCLIP in the xlarge-scale setting, which exhibits similar result, as shown in Appendix E.

In summary, the results in this section demonstrate the effectiveness of FastCLIP across different data scales (3 million to 315 million) and compute scales (1 to 8 nodes) in the limited-resource setting.

## 7 CONCLUSION

In this paper, we have proposed a distributed training framework of CLIP models in a resource-limited setting named FastCLIP. It leverages advanced compositional optimization with a novel gradient computation strategy to reduce the communication cost. We have investigated different optimization components, by proposing new techniques and benchmarking different techniques for each component under different settings to provide valuable insights on which techniques to use. Finally, leveraging the best-performant techniques from the experiment results, we compare the performance of FastCLIP with OpenCLIP on different data scales and compute scales, from 3 million to 315 million image-text pairs and from 1 node to 8 nodes. The results demonstrate that FastCLIP outperforms OpenCLIP by a large margin and achieves a significant speedup. This helps accelerate research in the areas of CLIP training and its various applications, as more researchers would be able to contribute their ideas and train CLIP models without access to a large amount of resources.

## 8 LIMITATIONS AND FUTURE WORK

Due to limited computing resources, we were unable to perform an extensive ablation study on the LAION315M dataset. As a future work, we will explore how to further improve the performance of FastCLIP in various aspects, e.g., reducing communication time and improving the convergence rate.

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

## A   DETAILS OF THE FASTCLIP FRAMEWORK

---

**Procedure 2:** contrastive_loss

---

```
/* global, individual τ:  temperature scheme (c.f.  Table 1) */
```
1 **if** *global $\tau$* **then**
2 $\quad$ Compute $g_{1,i}^t = g_1(\boldsymbol{w}^t, \tau^t, i, \mathcal{B}_{i-}^t), g_{2,i}^t = g_2(\boldsymbol{w}^t, \tau^t, i, \mathcal{B}_{i-}^t)$
3 **else if** *individual $\tau$* **then**
4 $\quad$ Compute $g_{1,i}^t = g_1(\boldsymbol{w}^t, \tau_{1,i}^t, i, \mathcal{B}_{i-}^t), g_{2,i}^t = g_2(\boldsymbol{w}^t, \tau_{2,i}^t, i, \mathcal{B}_{i-}^t)$

---

**Procedure 3:** gradient_estimator

---

```
/* global, individual τ:  temperature scheme (c.f.  Table 1) */
```
1 **if** *global $\tau$* **then**
2 $\quad$ Compute $G_{\boldsymbol{w},a,k}^t$ and $G_{\boldsymbol{w},b,k}^t$ using (2) and (3), respectively
3 **else if** *individual $\tau$* **then**
4 $\quad$ Compute $G_{\boldsymbol{w},a,k}^t$ and $G_{\boldsymbol{w},b,k}^t$ using (6) and (7), respectively

---

**Derivation of gradient of** (GCL) **w.r.t.** $\boldsymbol{w}$: Given a global batch $\mathcal{B}$, the gradient of (GCL) w.r.t. $\boldsymbol{w}$ is given by $G_{\boldsymbol{w},a} + G_{\boldsymbol{w},b}$, where

$$G_{\boldsymbol{w},a} = \tau \cdot \frac{1}{K} \sum_{k=1}^K \overbrace{\frac{1}{|\mathcal{B}_k|} \sum_{i \in \mathcal{B}_k} \frac{1}{\varepsilon + u_{1,i}} \cdot \frac{1}{K} \sum_{k'=1}^K \frac{1}{|\mathcal{B}_{k',i-}|} \sum_{j \in \mathcal{B}_{k',i-}} \nabla_1 \ell_1(\boldsymbol{e}_i, \boldsymbol{e}_{2,j}, \tau) \cdot \nabla \boldsymbol{e}_i}^{G_{\boldsymbol{w},a,1,k}}$$

$$+ \tau \cdot \frac{1}{K} \sum_{k=1}^K \overbrace{\frac{1}{|\mathcal{B}_k|} \sum_{i \in \mathcal{B}_k} \frac{1}{\varepsilon + u_{2,i}} \cdot \frac{1}{K} \sum_{k'=1}^K \frac{1}{|\mathcal{B}_{k',i-}|} \sum_{j \in \mathcal{B}_{k',i-}} \nabla_1 \ell_2(\boldsymbol{e}_i, \boldsymbol{e}_{1,j}, \tau) \cdot \nabla \boldsymbol{e}_i}^{G_{\boldsymbol{w},a,2,k}}.$$

$$G_{\boldsymbol{w},b} = \tau \cdot \frac{1}{K} \sum_{k=1}^K \frac{1}{|\mathcal{B}_k|} \sum_{i \in \mathcal{B}_k} \frac{1}{\varepsilon + u_{1,i}} \cdot \frac{1}{K} \sum_{k'=1}^K \frac{1}{|\mathcal{B}_{k',i-}|} \sum_{j \in \mathcal{B}_{k',i-}} \nabla_2 \ell_1(\boldsymbol{e}_i, \boldsymbol{e}_{2,j}, \tau) \cdot \nabla \boldsymbol{e}_{2,j}$$

$$+ \tau \cdot \frac{1}{K} \sum_{k=1}^K \frac{1}{|\mathcal{B}_k|} \sum_{i \in \mathcal{B}_k} \frac{1}{\varepsilon + u_{2,i}} \cdot \frac{1}{K} \sum_{k'=1}^K \frac{1}{|\mathcal{B}_{k',i-}|} \sum_{j \in \mathcal{B}_{k',i-}} \nabla_2 \ell_2(\boldsymbol{e}_i, \boldsymbol{e}_{1,j}, \tau) \cdot \nabla \boldsymbol{e}_{1,j}.$$

To compute $G_{\boldsymbol{w},a}$, we first gather all the $\boldsymbol{e}_{2,j}$ and $\boldsymbol{e}_{1,j}$ using ALL_GATHER to each worker, then compute $G_{\boldsymbol{w},a,1,k}$ and $G_{\boldsymbol{w},a,2,k}$ on the $k$-th worker, and average $G_{\boldsymbol{w},a,1,k}$ and $G_{\boldsymbol{w},a,2,k}$ over each

---

**Procedure 4:** parameter_update

---

**Input:** Parameter $\theta^t$ (can be $w$ or $\tau$) and its gradient estimator $G_\theta^t$, Weight decay $\lambda$, Learning rate $\eta_t$

**Optimizer** *SGD with momentum*

 **Additional Input :** Momentum parameter $\mu$

**1** Compute $m^{t+1} = \mu m^t + G_\theta^t + \lambda\theta^t$

**2** Set $\theta^{t+1} = \theta^t - \eta_t m^{t+1}$

**Optimizer** *LAMB*

 **Additional Input :** Hyperparameters $\beta_1, \beta_2, \epsilon$

**3** Compute $m^{t+1} = \beta_1 m^t + (1-\beta_1)G_\theta^t$

**4** Compute $v^{t+1} = \beta_2 v^t + (1-\beta_2)(G_\theta^t)^2$

**5** Compute $\hat{m}^{t+1} = m^{t+1}/(1-(\beta_1)^{t+1})$, $\hat{v}^{t+1} = v^{t+1}/(1-(\beta_2)^{t+1})$

**6** Compute $r^{t+1} = \hat{m}^{t+1}/(\sqrt{\hat{v}^{t+1}} + \epsilon)$

**7** **for** *each layer* $\theta^{t,(i)}$ *in* $\theta^t$ **do**

**8**  Compute $\alpha_{t,(i)} = \|\theta^{t,(i)}\|_2 / \|r^{t,(i)} + \lambda\theta^{t,(i)}\|_2$

**9**  Set $\theta^{t+1,(i)} = \theta^{t,(i)} - \eta_t \cdot \alpha_{t,(i)} \left(r^{t,(i)} + \lambda\theta^{t,(i)}\right)$

**Optimizer** *Lion*

 **Additional Input :** Hyperparameters $\beta_1, \beta_2$

**10** Compute $c^{t+1} = \beta_1 m^t + (1-\beta_1)G_\theta^t$

**11** Compute $m^{t+1} = \beta_2 m^t + (1-\beta_2)G_\theta^t$

**12** Set $\theta^{t+1} = \theta^t - \eta_t \left(\text{sign}(c^{t+1}) + \lambda\theta^t\right)$

**Optimizer** *AdamW*

 **Additional Input :** Hyperparameters $\beta_1, \beta_2, \epsilon$

**13** Compute $m^{t+1} = \beta_1 m^t + (1-\beta_1)G_\theta^t$

**14** Compute $v^{t+1} = \beta_2 v^t + (1-\beta_2)(G_\theta^t)^2$

**15** Compute $\hat{m}^{t+1} = m^{t+1}/(1-(\beta_1)^{t+1})$, $\hat{v}^{t+1} = v^{t+1}/(1-(\beta_2)^{t+1})$

**16** Set $\theta^{t+1} = \theta^t - \eta_t \left(\hat{m}^{t+1}/(\sqrt{\hat{v}^{t+1}} + \epsilon) + \lambda\theta^t\right)$

---

**Procedure 5:** temperature_update

---

**1** **if** *constant* $\tau$ **then**               `/* FastCLIP-v1 */`

**2** Set $\tau^{t+1} = \tau^t$

**3** **else if** *learnable* $\tau$ **then**

**4** **if** *loss is* (GCL) **then**             `/* FastCLIP-v0 */`

**5**  Compute $G_{\tau,k}^t$ using (8) and All_Reduce $G_\tau^t = \frac{1}{K}\sum_{l=1}^{K} G_{\tau,k}^t$

**6**  Update $\tau^{t+1}$ from $\tau^t$ and $G_\tau^t$ using Proc. 4 (with $\lambda = 0$)*

**7** **else if** *loss is* (RGCL) **then**          `/* FastCLIP-v2 */`

**8**  Compute $G_{\tau,1,i}^t, G_{\tau,2,i}^t$ for $i \in \mathcal{B}_k^t$ using (9)

**9**  Update $\tau_{1,i}^{t+1}$ from $\tau_{1,i}^t$ and $G_{\tau,1,i}^t$, and update $\tau_{2,i}^{t+1}$ from $\tau_{2,i}^t$ and $G_{\tau,2,i}^t$ using Proc. 4 (with $\lambda = 0$) for $i \in \mathcal{B}_k^t$

**10** **else if** *loss is* (RGCL-g) **then**        `/* FastCLIP-v3 */`

**11**  Compute $G_{\tau,k}^t$ using (10) and All_Reduce $G_\tau^t = \frac{1}{K}\sum_{l=1}^{K} G_{\tau,k}^t$

**12**  Update $\tau^{t+1}$ from $\tau^t$ and $G_\tau^t$ using Proc. 4 (with $\lambda = 0$)

---

*: Following OpenCLIP, we set the weight decay of the temperature parameter to 0.

worker using ALL_REDUCE. To compute $G_{w,b}$, we first switch the inner and outer averages:

$$G_{w,b} = \tau \cdot \frac{1}{K}\sum_{k'=1}^{K}\frac{1}{|\mathcal{B}_{k'}|}\sum_{j\in\mathcal{B}_{k'}} \cdot \overbrace{\frac{1}{K}\sum_{k=1}^{K}\frac{1}{|\mathcal{B}_{k,j-}|}\sum_{i\in\mathcal{B}_{k,j-}}\frac{1}{\varepsilon + u_{1,i}}\nabla_2\ell_1(e_i, e_{2,j}, \tau) \cdot \nabla e_{2,j}}^{G_{w,b,1,k'}}$$

$$+ \tau \cdot \frac{1}{K}\sum_{k'=1}^{K}\frac{1}{|\mathcal{B}_{k'}|}\sum_{j\in\mathcal{B}_{k'}} \cdot \overbrace{\frac{1}{K}\sum_{k=1}^{K}\frac{1}{|\mathcal{B}_{k,j-}|}\sum_{i\in\mathcal{B}_{k,j-}}\frac{1}{\varepsilon + u_{2,i}}\nabla_2\ell_2(e_i, e_{1,j}, \tau) \cdot \nabla e_{1,j}}^{G_{w,b,2,k'}}.$$

Then we gather all the $u_{1,i}$ and $u_{2,i}$ using ALL_GATHER to each worker, and compute $G_{\boldsymbol{w},b,1,k'}$ and $G_{\boldsymbol{w},b,2,k'}$ on the $k'$-th worker, then average $G_{\boldsymbol{w},b,1,k'}$ and $G_{\boldsymbol{w},b,2,k'}$ over each worker using ALL_REDUCE to get $G_{\boldsymbol{w},b}$. For practical consideration, we switch the inner and outer averages in $G_{\boldsymbol{w},b,1,k'}$ and $G_{\boldsymbol{w},b,2,k'}$ again so that we can compute them along with $G_{\boldsymbol{w},a,1,k}$ and $G_{\boldsymbol{w},a,2,k}$ using the same function:

$$
\begin{aligned}
G_{\boldsymbol{w},b,1,k'} &= \frac{1}{|\mathcal{B}_{k'}|} \sum_{j \in \mathcal{B}_{k'}} \cdot \frac{1}{K} \sum_{k=1}^{K} \frac{1}{|\mathcal{B}_{k,j-}|} \sum_{i \in \mathcal{B}_{k,j-}} \frac{1}{\varepsilon + u_{1,i}} \nabla_2 \ell_1(\boldsymbol{e}_i, \boldsymbol{e}_{2,j}, \tau) \cdot \nabla \boldsymbol{e}_{2,j} \\
&\stackrel{(*)}{=} \frac{1}{|\mathcal{B}_{k'}|} \sum_{j \in \mathcal{B}_{k'}} \cdot \frac{1}{|\mathcal{B}_{j-}|} \sum_{i \in \mathcal{B}_{j-}} \frac{1}{\varepsilon + u_{1,i}} \nabla_2 \ell_1(\boldsymbol{e}_i, \boldsymbol{e}_{2,j}, \tau) \cdot \nabla \boldsymbol{e}_{2,j} \\
&= \frac{1}{|\mathcal{B}|} \sum_{i \in \mathcal{B}} \frac{1}{\varepsilon + u_{1,i}} \cdot \frac{1}{|\mathcal{B}_{k'}|} \cdot \frac{|\mathcal{B}|}{|\mathcal{B}_{j-}|} \sum_{j \in \mathcal{B}_{k',i-}} \nabla_2 \ell_1(\boldsymbol{e}_i, \boldsymbol{e}_{2,j}, \tau) \cdot \nabla \boldsymbol{e}_{2,j},
\end{aligned}
$$

where $(*)$ uses the fact that the average over local batch and workers is equal to the average over the global batch. Similarly,

$$
G_{\boldsymbol{w},b,2,k'} = \frac{1}{|\mathcal{B}|} \sum_{i \in \mathcal{B}} \frac{1}{\varepsilon + u_{2,i}} \cdot \frac{1}{|\mathcal{B}_{k'}|} \cdot \frac{|\mathcal{B}|}{|\mathcal{B}_{j-}|} \sum_{j \in \mathcal{B}_{k',i-}} \nabla_2 \ell_2(\boldsymbol{e}_i, \boldsymbol{e}_{1,j}, \tau) \cdot \nabla \boldsymbol{e}_{1,j}.
$$

**Deferred Computation in Alg.1**: At iteration $t$, for SogCLR and other algorithms with global temperature parameter (except FastCLIP-v0), the gradient estimator for $\boldsymbol{w}$ on $k$-th worker is computed as

$$
\begin{aligned}
G_{\boldsymbol{w},a,k}^t = \frac{\tau^t}{|\mathcal{B}_k^t|} \sum_{i \in \mathcal{B}_k^t} \Bigg( &\frac{1}{\varepsilon + u_{1,i}^{t+1}} \left( \frac{1}{|\mathcal{B}_{i-}^t|} \sum_{j \in \mathcal{B}_{i-}^t} \nabla_1 \ell_1(\boldsymbol{e}_i, \boldsymbol{e}_{2,j}, \tau^t) \cdot \nabla \boldsymbol{e}_i \right) \\
&+ \frac{1}{\varepsilon + u_{2,i}^{t+1}} \left( \frac{1}{|\mathcal{B}_{i-}^t|} \sum_{j \in \mathcal{B}_{i-}^t} \nabla_1 \ell_2(\boldsymbol{e}_i, \boldsymbol{e}_{1,j}, \tau^t) \cdot \nabla \boldsymbol{e}_i \right) \Bigg).
\end{aligned} \tag{2}
$$

$$
\begin{aligned}
G_{\boldsymbol{w},b,k}^t = \frac{\tau^t}{|\mathcal{B}^t|} \sum_{i \in \mathcal{B}^t} \Bigg( &\frac{1}{\varepsilon + u_{1,i}^{t+1}} \left( \frac{1}{|\mathcal{B}_k^t|} \cdot \frac{|\mathcal{B}^t|}{|\mathcal{B}_{i-}^t|} \sum_{j \in \mathcal{B}_{k,i-}^t} \nabla_2 \ell_1(\boldsymbol{e}_i, \boldsymbol{e}_{2,j}, \tau^t) \cdot \nabla \boldsymbol{e}_{2,j} \right) \\
&+ \frac{1}{\varepsilon + u_{2,i}^{t+1}} \left( \frac{1}{|\mathcal{B}_k^t|} \cdot \frac{|\mathcal{B}^t|}{|\mathcal{B}_{i-}^t|} \sum_{j \in \mathcal{B}_{k,i-}^t} \nabla_2 \ell_2(\boldsymbol{e}_i, \boldsymbol{e}_{1,j}, \tau^t) \cdot \nabla \boldsymbol{e}_{1,j} \right) \Bigg).
\end{aligned} \tag{3}
$$

For FastCLIP-v0, we need to remove the $\tau^t$ at the front:

$$
\begin{aligned}
G_{\boldsymbol{w},a,k}^t = \frac{1}{|\mathcal{B}_k^t|} \sum_{i \in \mathcal{B}_k^t} \Bigg( &\frac{1}{\varepsilon + u_{1,i}^{t+1}} \left( \frac{1}{|\mathcal{B}_{i-}^t|} \sum_{j \in \mathcal{B}_{i-}^t} \nabla_1 \ell_1(\boldsymbol{e}_i, \boldsymbol{e}_{2,j}, \tau^t) \cdot \nabla \boldsymbol{e}_i \right) \\
&+ \frac{1}{\varepsilon + u_{2,i}^{t+1}} \left( \frac{1}{|\mathcal{B}_{i-}^t|} \sum_{j \in \mathcal{B}_{i-}^t} \nabla_1 \ell_2(\boldsymbol{e}_i, \boldsymbol{e}_{1,j}, \tau^t) \cdot \nabla \boldsymbol{e}_i \right) \Bigg).
\end{aligned} \tag{4}
$$

$$
\begin{aligned}
G_{\boldsymbol{w},b,k}^t = \frac{1}{|\mathcal{B}^t|} \sum_{i \in \mathcal{B}^t} \Bigg( &\frac{1}{\varepsilon + u_{1,i}^{t+1}} \left( \frac{1}{|\mathcal{B}_k^t|} \cdot \frac{|\mathcal{B}^t|}{|\mathcal{B}_{i-}^t|} \sum_{j \in \mathcal{B}_{k,i-}^t} \nabla_2 \ell_1(\boldsymbol{e}_i, \boldsymbol{e}_{2,j}, \tau^t) \cdot \nabla \boldsymbol{e}_{2,j} \right) \\
&+ \frac{1}{\varepsilon + u_{2,i}^{t+1}} \left( \frac{1}{|\mathcal{B}_k^t|} \cdot \frac{|\mathcal{B}^t|}{|\mathcal{B}_{i-}^t|} \sum_{j \in \mathcal{B}_{k,i-}^t} \nabla_2 \ell_2(\boldsymbol{e}_i, \boldsymbol{e}_{1,j}, \tau^t) \cdot \nabla \boldsymbol{e}_{1,j} \right) \Bigg).
\end{aligned} \tag{5}
$$

For iSogCLR and other algorithms with individual temperature parameter, it is computed using a slightly different formula (the $\tau$ part is different)

$$
G_{\boldsymbol{w},a,k}^t = \frac{1}{|\mathcal{B}_k^t|} \sum_{i \in \mathcal{B}_k^t} \left( \frac{\tau_{1,i}^t}{\varepsilon + u_{1,i}^{t+1}} \left( \frac{1}{|\mathcal{B}_{i-}^t|} \sum_{j \in \mathcal{B}_{i-}^t} \nabla_1 \ell_1(\boldsymbol{e}_i, \boldsymbol{e}_{2,j}, \tau_{1,i}^t) \cdot \nabla \boldsymbol{e}_i \right) \right.
$$
$$
\left. + \frac{\tau_{2,i}^t}{\varepsilon + u_{2,i}^{t+1}} \left( \frac{1}{|\mathcal{B}_{i-}^t|} \sum_{j \in \mathcal{B}_{i-}^t} \nabla_1 \ell_2(\boldsymbol{e}_i, \boldsymbol{e}_{1,j}, \tau_{2,i}^t) \cdot \nabla \boldsymbol{e}_i \right) \right). \tag{6}
$$

$$
G_{\boldsymbol{w},b,k}^t = \frac{1}{|\mathcal{B}^t|} \sum_{i \in \mathcal{B}^t} \left( \frac{\tau_{1,i}^t}{\varepsilon + u_{1,i}^{t+1}} \left( \frac{1}{|\mathcal{B}_k^t|} \cdot \frac{|\mathcal{B}^t|}{|\mathcal{B}_{i-}^t|} \sum_{j \in \mathcal{B}_{k,i-}^t} \nabla_2 \ell_1(\boldsymbol{e}_i, \boldsymbol{e}_{2,j}, \tau_{1,i}^t) \cdot \nabla \boldsymbol{e}_{2,j} \right) \right.
$$
$$
\left. + \frac{\tau_{2,i}^t}{\varepsilon + u_{2,i}^{t+1}} \left( \frac{1}{|\mathcal{B}_k^t|} \cdot \frac{|\mathcal{B}^t|}{|\mathcal{B}_{i-}^t|} \sum_{j \in \mathcal{B}_{k,i-}^t} \nabla_2 \ell_2(\boldsymbol{e}_i, \boldsymbol{e}_{1,j}, \tau_{2,i}^t) \cdot \nabla \boldsymbol{e}_{1,j} \right) \right). \tag{7}
$$

FastCLIP-v0 computes the following gradient estimator for $\tau$:

$$
G_{\tau,k}^t = \frac{1}{|\mathcal{B}_k^t|} \sum_{i \in \mathcal{B}_k^t} \frac{1}{\varepsilon + u_{1,i}^{t+1}} \cdot \frac{1}{|\mathcal{B}_{i-}^t|} \sum_{j \in \mathcal{B}_{i-}^t} \nabla_3 \ell_1(\boldsymbol{e}_i, \boldsymbol{e}_{2,j}, \tau^t),
$$
$$
+ \frac{1}{|\mathcal{B}_k^t|} \sum_{i \in \mathcal{B}_k^t} \frac{1}{\varepsilon + u_{2,i}^{t+1}} \cdot \frac{1}{|\mathcal{B}_{i-}^t|} \sum_{j \in \mathcal{B}_{i-}^t} \nabla_3 \ell_2(\boldsymbol{e}_i, \boldsymbol{e}_{1,j}, \tau^t). \tag{8}
$$

FastCLIP-v2 computes the following gradient estimators for $\tau$:

$$
G_{\tau,1,i}^t = \frac{1}{|\mathcal{S}|} \left( \log\left(\varepsilon + u_{1,i}^{t+1}\right) + \rho + \tau_{1,i}^t \cdot \frac{1}{\varepsilon + u_{1,i}^{t+1}} \cdot \frac{1}{|\mathcal{B}_{i-}^t|} \sum_{j \in \mathcal{B}_{i-}^t} \nabla_3 \ell_1(\boldsymbol{e}_i, \boldsymbol{e}_{2,j}, \tau_{1,i}^t) \right),
$$
$$
G_{\tau,2,i}^t = \frac{1}{|\mathcal{S}|} \left( \log\left(\varepsilon + u_{2,i}^{t+1}\right) + \rho + \tau_{2,i}^t \cdot \frac{1}{\varepsilon + u_{2,i}^{t+1}} \cdot \frac{1}{|\mathcal{B}_{i-}^t|} \sum_{j \in \mathcal{B}_{i-}^t} \nabla_3 \ell_2(\boldsymbol{e}_i, \boldsymbol{e}_{1,j}, \tau_{2,i}^t) \right), \tag{9}
$$

FastCLIP-v3 computes the following gradient estimator for $\tau$:

$$
G_{\tau,k}^t = \frac{1}{|\mathcal{B}_k^t|} \sum_{i \in \mathcal{B}_k^t} \left( \log\left(\varepsilon + u_{1,i}^{t+1}\right) + \log\left(\varepsilon + u_{2,i}^{t+1}\right) \right) + 2\rho
$$
$$
+ \tau^t \cdot \frac{1}{|\mathcal{B}_k^t|} \sum_{i \in \mathcal{B}_k^t} \frac{1}{\varepsilon + u_{1,i}^{t+1}} \cdot \frac{1}{|\mathcal{B}_{i-}^t|} \sum_{j \in \mathcal{B}_{i-}^t} \nabla_3 \ell_1(\boldsymbol{e}_i, \boldsymbol{e}_{2,j}, \tau^t) \tag{10}
$$
$$
+ \tau^t \cdot \frac{1}{|\mathcal{B}_k^t|} \sum_{i \in \mathcal{B}_k^t} \frac{1}{\varepsilon + u_{2,i}^{t+1}} \cdot \frac{1}{|\mathcal{B}_{i-}^t|} \sum_{j \in \mathcal{B}_{i-}^t} \nabla_3 \ell_2(\boldsymbol{e}_i, \boldsymbol{e}_{1,j}, \tau^t).
$$

## B    EXPERIMENT HYPERPARAMETERS

Unless otherwise specified, for both FastCLIP and OpenCLIP, we use AdamW as the optimizer. For all settings, we use a cosine learning rate (LR) schedule for updating model parameters, which first linearly increases the LR from 0 to peak LR in the warmup stage, then decreases it following a cosine function. The hyperparameters we use are specified in Table 7. Other hyperparameters regarding the inner learning rate schedule, temperature parameter updates, and the LAMB optimizer will be introduced in the paragraphs that follow.

Table 7: Hyperparameters for different settings. $\beta_1, \beta_2, \epsilon$ are hyperparameters in the AdamW optimizer. lr denotes the peak learning rate. min_lr denotes the learning rate at the end of training. wd denotes the weight decay. warmup denotes the number of iterations in the warmup stage.

| Setting | $\beta_1$ | $\beta_2$ | $\epsilon$ | lr | min_lr | wd | warmup |
|---------|-----------|-----------|------------|------|--------|-----|--------|
| Medium | 0.9 | 0.999 | 1e-8 | 1e-3 | 0 | 0.1 | 10k |
| Large | 0.9 | 0.98 | 1e-6 | 4e-4 | 0 | 0.1 | 10k |
| xLarge | 0.9 | 0.98 | 1e-6 | 2e-4 | 0 | 0.2 | 13k |

**Experiments benchmarking the inner LR schedule**: We compare three pairs of approaches: SogCLR and FastCLIP-v1; iSogCLR and FastCLIP-v2; FastCLIP-v3 with constant $\gamma$ and FastCLIP-v3, where the former of each pair uses constant $\gamma$ schedule and the latter uses cosine $\gamma$ schedule. Any two approaches of each pair only differ in $\gamma$ schedule. For approaches using constant $\gamma$ schedule, we tune the value of $\gamma$ in $\{0.2, 0.4, 0.6, 0.8\}$. For approaches using cosine $\gamma$ schedule, we tune the value of $\gamma_{\min}$ (the value $\gamma$ will decay to in the end) in $\{0.2, 0.6\}$ and decay epochs in $\{50\%, 100\%\}$ of the number of training epochs. The $\gamma$ values for each algorithm are presented in Table 8. Other hyperparameters are kept the same within each pair. For SogCLR and FastCLIP-v1, we set the temperature parameter to 0.03. For iSogCLR and FastCLIP-v2, we set the initial temperature parameter to 0.03, $\rho$ to 9.0, and the learning rate of $\tau$ to 1e-2. For FastCLIP-v3 with constant $\gamma$ schedule and FastCLIP-v3, we set the initial temperature parameter to 0.07, $\rho$ to 6.5 in the medium-scale setting and 8.5 in the large-scale setting, and learning rate of $\tau$ to 2e-4 in the medium-scale setting and 1e-4 in the large-scale setting. For FastCLIP-v3, its learning rate of $\tau$ decays to 1/3 of its original value when $\tau$ becomes smaller than 0.03.

Table 8: Values of $\gamma$ for different schedules in different settings. For Cosine $\gamma$ schedule, we report the $\gamma$ value along with number of $\gamma$ decay epochs $E$ (c.f. Section 5). *: v3 (Const. $\gamma$) denotes FastCLIP-v3 with constant $\gamma$ schedule.

| Setting | Constant $\gamma$ | | Cosine $\gamma$ | |
|---------|-----------|------|-----------|------|
| | Algorithm | $\gamma$ | Algorithm | $\gamma_{\min}, E$ |
| Medium | SogCLR | 0.6 | FastCLIP-v1 | 0.2, 18 |
| | iSogCLR | 0.6 | FastCLIP-v2 | 0.2, 18 |
| | v3 (Const. $\gamma$)* | 0.6 | FastCLIP-v3 | 0.2, 18 |
| Large | SogCLR | 0.6 | FastCLIP-v1 | 0.2, 16 |
| | iSogCLR | 0.8 | FastCLIP-v2 | 0.6, 16 |
| | v3 (Const. $\gamma$)* | 0.6 | FastCLIP-v3 | 0.2, 16 |
| xLarge | - | - | FastCLIP-v3 | 0.8, 10 |

**Experiments benchmarking the temperature parameter updates**: For all algorithms we leverage a cosine $\gamma$ schedule with $\gamma_{\min} = 0.2$ and decay epochs $E$ equal to 50% of the number of training epochs. For all algorithms, we tune their initial temperature parameter in $\{0.03, 0.05, 0.07\}$. For FastCLIP-v2 and -v3, we tune $\rho$ in $[6.0, 9.0]$, we also tune the learning rate of $\tau$ in $[1e-4, 1e-2]$. Other hyperparameters are kept the same for the four algorithms. The tuned initial temperature is 0.07 for FastCLIP-v3 and 0.03 for other algorithms. The $\rho$ values are presented in Table 9. For FastCLIP-v2, the tuned learning rate of $\tau$ is 1e-2 in the medium-scale setting and 1e-4 in the large-scale setting. For FastCLIP-v3, the tuned learning rate of $\tau$ is 2e-4 in the medium-scale setting and 1e-4 in the large-scale setting. For FastCLIP-v3, its learning rate of $\tau$ decays to 1/3 of its original value when $\tau$ becomes smaller than 0.03.

Table 9: Value of $\rho$ for FastCLIP-v2 and -v3 in different settings.

| Algorithm | Medium | Large | xLarge |
|-----------|--------|-------|--------|
| FastCLIP-v2 | 7.0 | 8.5 | - |
| FastCLIP-v3 | 6.5 | 8.5 | 16.0 |

**Experiments benchmarking the optimizer**: We use FastCLIP-v3 as the base algorithm. For SGD with momentum, we tune its learning rate of model parameters in [4e-5, 4e0] and weight decay in [1e-6, 0.2]. For all other optimizers, we tune their learning rate of model parameters in [4e-5, 4e-3] and weight decay in [0.01, 0.2]. Other hyperparameters are kept the same as in Temperature Parameter Updates. The tuned learning rate of model parameters and weight decay are reported in Table 10. Following OpenCLIP (Cherti et al., 2023), we set the weight decay of the temperature parameter to 0. And following EVA-CLIP (Sun et al., 2023) in the implementation of LAMB, we set $\alpha$ at Line 9 in Proc. 4 to 1.0 when updating the temperature parameter, leading to the same update as AdamW.

Table 10: Values of learning rate of model parameters and weight decay for different optimizers. SGDM denotes SGD with momentum.

| Hyperparameters | Medium | | | | Large | | | |
|---|---|---|---|---|---|---|---|---|
| | SGDM | LAMB | Lion | AdamW | SGDM | LAMB | Lion | AdamW |
| Learning rate | 1.0 | 2e-3 | 2e-4 | 1e-3 | 2.0 | 2e-3 | 1e-4 | 4e-4 |
| Weight decay | 3e-6 | 0.1 | 0.3 | 0.1 | 3e-6 | 0.1 | 0.3 | 0.1 |

**Experiments demonstrating the scaling performance**: We tune the learning rate of model parameters of OpenCLIP on 2 nodes in the medium-scale and large-scale setting in $[4e - 5, 4e - 3]$, and on 4 nodes in the xlarge-scale setting in $[4e - 5, 4e - 4]$. The tuned learning rate of model parameters of OpenCLIP is 1e-3, 4e-4 and 2e-4 in the medium-scale, large-scale and xlarge-scale setting, respectively. Other hyperparameters are set according to Table 7 to 9. In the xlarge-scale setting, we set the learning rate of model parameters of FastCLIP-v3 to the same value as OpenCLIP. For different number of nodes in the medium-scale and large-scale setting, we scale the learning rate of model parameters and temperature parameter linearly in proportion to global batch size and keep other hyperparameters unchanged. For FastCLIP-v3 in the xlarge-scale setting, we set $\rho$ to 16.0 and the learning rate of temperature parameter to 5e-5. We leverage a cosine $\gamma$ schedule with $\gamma_{\min} = 0.8$ and decay epochs $E = 10$.

**Choice of $\gamma_{\min}$ in the xlarge-scale setting**: Note that in the xlarge-scale setting we use a larger $\gamma_{\min}$ value than in the medium-scale and large-scale settings. We find that the batch size impacts how we should set the $\gamma_{\min}$ value. To illustrate this, we conduct two sets of experiments in the large-scale setting on 2 nodes and 8 nodes, respectively. Each set is FastCLIP-v3 with different $\gamma_{\min}$ value. The results are plotted in Figure 5. Comparing a larger $\gamma_{\min}$ (0.8) with a smaller one (0.2) in the same setting, we find that the training can be split into three stages. In the first stage, the two runs have similar performance. In the second stage, larger $\gamma_{\min}$ outperforms the smaller one, while the smaller one catches up with the larger one and outperforms it in the last stage. From Figure 5 we can also observe that with a larger global batch size, the second stage becomes longer. Note that in the medium-scale and large-scale settings we use a global batch size of 1024 and 2048 respectively, while we set it to 5120 in the xlarge-scale setting. We also conjecture that the second stage becomes longer as the data scales up, though we did not validate this due to resource limits. The large batch size and large data scale in the xlarge-scale setting motivate our use of a larger $\gamma_{\min}$ value than in the medium-scale and large-scale settings.

## C  THE IMPACT OF BATCH SIZE AND DATASET SIZE ON OPENCLIP

The ImageNet-1k top 1 accuracy of OpenCLIP in the xlarge-scale setting (LAION315M for 13B samples, batch size 5120) is 62.90%, while the result of OpenCLIP reported in Cherti et al. (2023) (LAION400M for 13B samples, batch size 33792) is 67.00%. We attribute the gap to smaller batch size and smaller dataset size. We first summarize some existing results that demonstrate the impact of these two factors:

- Batch size: We use a smaller batch size of 5120 for the xlarge scale training due to limited compute resources, which is 6 times smaller than the batch size used in Cherti et al. (2023) (33792, with 67% performance of ViT-B/16) and 12.8 times smaller than that in Chen et al. (2023b) (65536, with 64.3% performance of ViT-B/32). As reported in existing works, e.g., Chen et al. (2023b), batch size has an important impact on OpenCLIP. The results in the table above clearly demonstrate

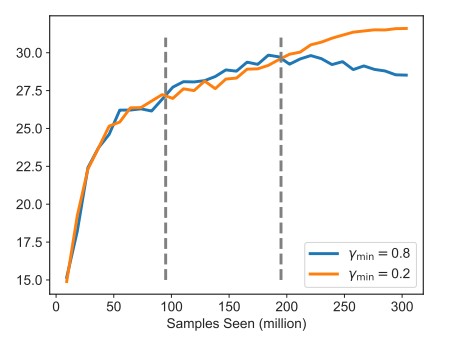 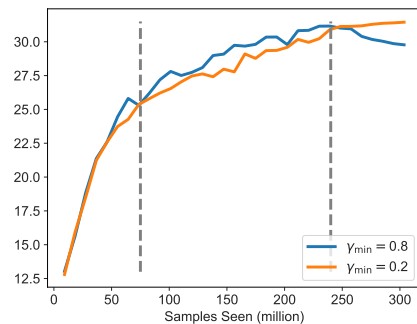

(a) 2 Nodes, Batch size 2048          (b) 8 Nodes, Batch size 8192

Figure 5: Datacomp average performance of FastCLIP-v3 with $\gamma$ decay epochs 16 (145 million samples seen) and different $\gamma_{\min}$ in the large-scale setting. Batch size denotes global batch size. The vertical dashed lines divided the plot into three parts (c.f. Choice of $\gamma_{\min}$ in the xlarge-scale Setting in Appendix B).

Table 11: Summary of existing results of training using OpenCLIP.

| Work | Architecture | Data Size (M) | Batch Size | Samples (B) | Performance (%) |
|------|--------------|---------------|------------|-------------|-----------------|
| Cherti et al. (2023) | ViT-B/16 | 80 | 90112 | 13 | 60.24 |
| Cherti et al. (2023) | ViT-B/16 | 400 | 33792 | 13 | 67.00 |
| Cherti et al. (2023) | ViT-B/16 | 2000 | 90112 | 13 | 68.13 |
| Chen et al. (2023b) | ViT-B/32 | 100 | 8192 | 1.6 | 48.76 |
| Chen et al. (2023b) | ViT-B/32 | 100 | 16384 | 1.6 | 50.95 |
| Chen et al. (2023b) | ViT-B/32 | 100 | 32768 | 1.6 | 51.64 |
| Chen et al. (2023b) | ViT-B/32 | 100 | 65536 | 1.6 | 51.91 |
| Chen et al. (2023b) | ViT-B/32 | 400 | 65536 | 13 | 64.3 |
| OpenCLIP (our impl.) | ViT-B/16 | 315 | 5120 | 13 | 62.90 |

this. If we fit the performance in Chen et al. (2023b) for different batch sizes (rows 4-7 in the table above) with a reciprocal function $p = -a/x + b$, where $x$ is the batch size and $p$ is the ImageNet-1k top 1 accuracy, the results (plotted in Figure 6 (a)) showed that the predicted performance with batch size 5120 has a 5% drop compared with batch size 32768. This is somewhat consistent with that our result using 5120 batch size has a 4.1% drop in performance for OpenCLIP than using 33792 batch size as in Cherti et al. (2023).

- Dataset size: Although we intended to use LIAON400M, due to broken URLs we could only download a subset of the LAION400M dataset, which consists of 315M image-text pairs. This is also a factor contributing to the worse performance of OpenCLIP as Cherti et al. (2023) reported that using 80M data leads to a performance drop by 7% compared with 400M data (rows 1-2 in the table above). If we fit the results in Cherti et al. (2023) for different data sizes with a power function $p = \alpha x^{\beta} + p_0$, where $x$ is the dataset size and $p$ is the ImageNet-1k top 1 accuracy. The results (plotted in Figure 6 (b)) showed that the predicted performance of OpenCLIP training ViT-B/16 on a 315M dataset with 13B samples seen and at least 33K batch size is 64.5%. Our OpenCLIP using a smaller batch size of 5120 (last row of the table) achieves 62.90%, which is expected considering the small batch size.

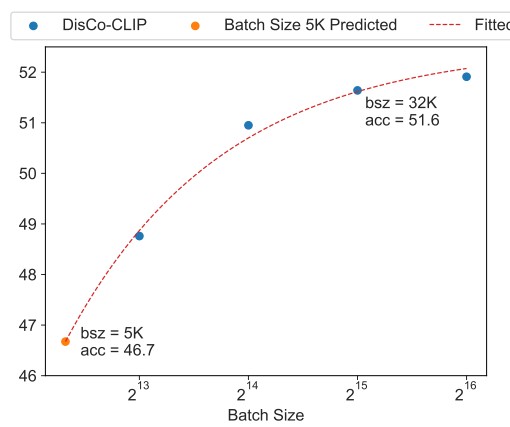
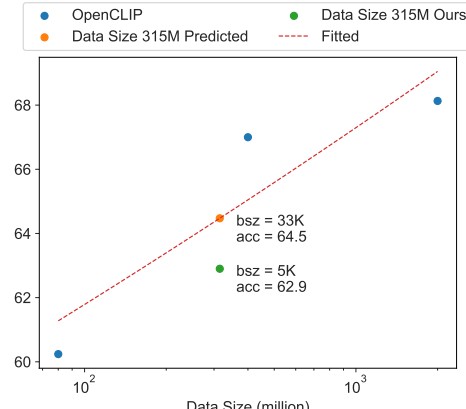

(a) Results of Chen et al. (2023b). Blue dots: results from Chen et al. (2023b). Red line: fitted using blue dots. Orange dot: predicted result.

(b) Results of Cherti et al. (2023). Blue dots: results from Cherti et al. (2023). Red line: fitted using blue dots. Orange dot: predicted result. Green dot: our result.

Figure 6: ImageNet-1k top 1 accuracy plots. 'bsz' denotes batch size and 'acc' denotes accuracy.

## D    THE EFFECT OF $\varepsilon$ IN (RGCL-g)

We found that in the xlarge-scale setting, the constant $\varepsilon$ plays an important role in the performance of FastCLIP-v3. In particular, the gradient estimators $G^t_{\boldsymbol{w},a,k}$ and $G^t_{\boldsymbol{w},b,k}$ of (RGCL-g) in Equation (2) and (3) are scaled by two factors: $1/(\varepsilon + u^{t+1}_{1,i})$ and $1/(\varepsilon + u^{t+1}_{2,i})$. Recall that $\boldsymbol{u}_{1,i}$ and $\boldsymbol{u}_{2,i}$ are approximations of $g_1(\boldsymbol{w}^t, \tau^t, i, \mathcal{S}_{i-})$ and $g_2(\boldsymbol{w}^t, \tau^t, i, \mathcal{S}_{i-})$, respectively. Thus, in the later stage of training many examples (those that are well-learned) will have very small $\boldsymbol{u}^{t+1}_{1,i}$ and $\boldsymbol{u}^{t+1}_{2,i}$. Then with a very small $\varepsilon$ the scaling factors in the estimated gradient for these samples will be very large, which may suffer from over-optimization for those examples and harm generalization. In the following figure, we plot the performance of FastCLIP-v3 with two schemes of $\varepsilon$ along with the performance of OpenCLIP (in blue): i) $\varepsilon=$ 1e-14 (in orange) and ii) $\varepsilon=$ 1e-6 (in green). The value 1e-6 is not tuned due to limited compute resources and the three experiments were run for only 30 epochs (9.45B samples seen). From Figure 7 we can see that with larger $\varepsilon$, both the ImageNet-1k top 1 accuracy and Datacomp Average performance improve by a large margin.

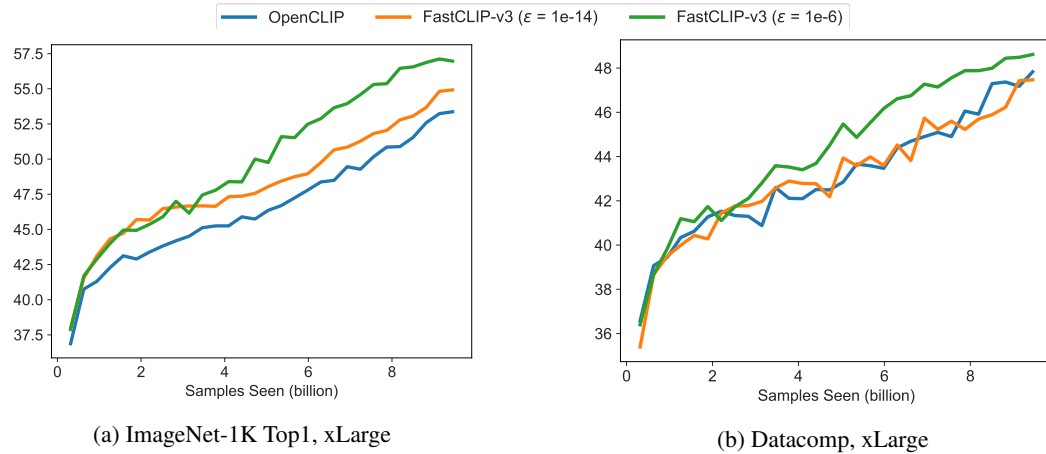

(a) ImageNet-1K Top1, xLarge

(b) Datacomp, xLarge

Figure 7: ImageNet-1k Top 1 accuracy (left) and Datacomp Average performance (right) of FastCLIP-v3 with different $\varepsilon$ in the xlarge-scale setting.

# E  MORE EXPERIMENT RESULTS

## E.1  OPTIMIZATION COMPONENTS

We plot the Datacomp average performance curves of different algorithms with constant $\gamma$ schedule and cosine $\gamma$ schedule in Figure 8, which corresponds to Table 3 in Section 5. We plot the Datacomp average performance curves of algorithms with different temperature updates in Figure 9 (a) and (b), which corresponds to Table 4 in Section 5. We plot the Datacomp average performance curves of FastCLIP-v3 with AdamW and LAMB optimizer in Figure 9 (c) and (d), which corresponds to Table 5 in Section 5.

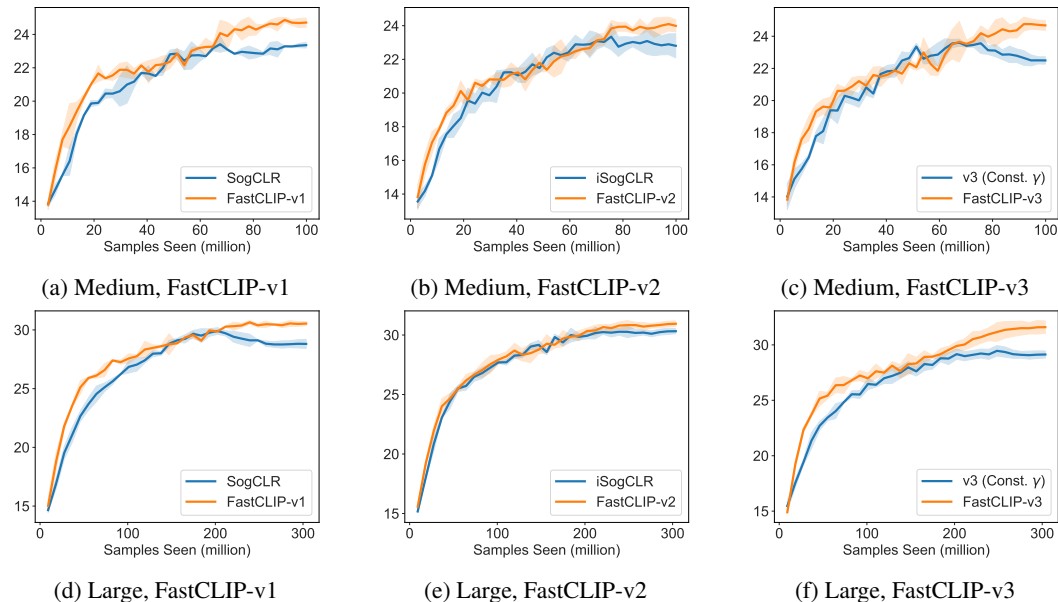

Figure 8: Datacomp performance of algorithms with constant $\gamma$ schedule and cosine $\gamma$ schedule. v3 (Const. $\gamma$) denotes FastCLIP-v3 with constant $\gamma$ schedule.

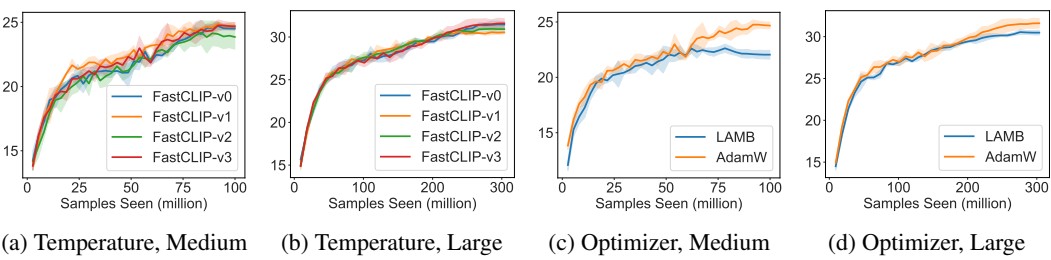

Figure 9: Subfigures (a), (b) present the Datacomp performance of algorithms with different temperature parameter updates in the medium-scale and large-scale setting, respectively. Subfigures (c), (d) present the Datacomp performance of FastCLIP-v3 with different optimizers in the medium-scale and large-scale setting, respectively.

## E.2  SCALING PERFORMANCE

In this subsection we provide more results to complement the figures in Section 6.

**Performance of OpenCLIP and FastCLIP-v3**: The data to plot Figure 2 is presented in Table 13 and Table 14. We also provide the Datacomp performance in Table 12. The Datacomp performance of OpenCLIP and FastCLIP-v3 in the xlarge-scale setting is plotted in Figure 10. In the xlarge-scale

setting, we also conduct experiments of FastCLIP-v3 on a 1.4B subset of the DFN-2B dataset (Fang et al., 2023) that originally includes 1.9B image-text pairs. The ImageNet-1K top 1 accuracy and Datacomp Average performance of FastCLIP-v3 on different datasets are shown in Figure 11. We can see that our approach is still effective on extremely large-scale data.

Table 12: Datacomp Average performance of OpenCLIP and FastCLIP-v3 trained on different number of nodes. Improvement denotes the absolute difference between FastCLIP-v3 and OpenCLIP.

| Setting | Algorithm | 1 Node | 2 Nodes | 4 Nodes | 8 Nodes |
|---------|-----------|--------|---------|---------|---------|
| Medium | OpenCLIP | 21.82 (0.59) | 21.84 (0.23) | 21.65 (0.13) | 22.22 (0.37) |
| | FastCLIP-v3 | 24.54 (0.25) | 24.76 (0.26) | 24.43 (0.20) | 25.23 (0.28) |
| | *Improvement* | *2.72* | *2.92* | *2.78* | *3.01* |
| Large | OpenCLIP | 27.55 (0.46) | 27.91 (0.73) | 28.93 (0.29) | 28.75 (0.59) |
| | FastCLIP-v3 | 30.81 (0.38) | 31.60 (0.46) | 31.65 (0.13) | 31.45 (0.32) |
| | *Improvement* | *3.26* | *3.69* | *2.72* | *2.70* |

Table 13: Retrieval performance of OpenCLIP and FastCLIP-v3 trained on different number of nodes. Improvement denotes the absolute difference between FastCLIP-v3 and OpenCLIP.

| Setting | Algorithm | 1 Node | 2 Nodes | 4 Nodes | 8 Nodes |
|---------|-----------|--------|---------|---------|---------|
| Medium | OpenCLIP | 24.07 (0.16) | 25.20 (0.22) | 25.07 (0.26) | 26.20 (0.10) |
| | FastCLIP-v3 | 30.02 (0.57) | 30.36 (0.18) | 30.42 (0.24) | 30.42 (0.24) |
| | *Improvement* | *5.95* | *5.16* | *5.35* | *4.22* |
| Large | OpenCLIP | 29.17 (0.17) | 29.58 (0.62) | 30.25 (0.31) | 30.87 (0.11) |
| | FastCLIP-v3 | 33.90 (0.28) | 34.88 (0.28) | 34.91 (0.16) | 34.74 (0.31) |
| | *Improvement* | *4.73* | *5.30* | *4.66* | *3.87* |

Table 14: ImageNet & Variants accuracy of OpenCLIP and FastCLIP-v3 trained on different number of nodes. Improvement denotes the absolute difference between FastCLIP-v3 and OpenCLIP.

| Setting | Algorithm | 1 Node | 2 Nodes | 4 Nodes | 8 Nodes |
|---------|-----------|--------|---------|---------|---------|
| Medium | OpenCLIP | 14.16 (0.11) | 14.73 (0.22) | 15.24 (0.26) | 16.03 (0.23) |
| | FastCLIP-v3 | 18.37 (0.26) | 19.08 (0.16) | 19.21 (0.18) | 19.20 (0.16) |
| | *Improvement* | *4.21* | *4.35* | *3.97* | *3.17* |
| Large | OpenCLIP | 20.51 (0.14) | 21.08 (0.09) | 22.32 (0.23) | 22.77 (0.14) |
| | FastCLIP-v3 | 23.76 (0.38) | 24.78 (0.28) | 24.79 (0.20) | 24.93 (0.16) |
| | *Improvement* | *3.25* | *3.70* | *2.47* | *2.16* |

**Training Time Comparison between OpenCLIP and FastCLIP-v3**: We present the training time breakdown of OpenCLIP and FastCLIP-v3 in Table 15 and 16 for the medium-scale and large-scale settings, respectively. We can see that as the number of nodes scales up, the computation time of OpenCLIP and FastCLIP-v3 is always close to each other, while the gap in communication time becomes much larger, which is also depicted in subfigures (c) and (d). Even if we exclude the part of communication that overlaps with computation, the gap in pure communication still becomes larger with increasing number of nodes, and thus FastCLIP-v3 has a shorter running time on 4 and 8 nodes.

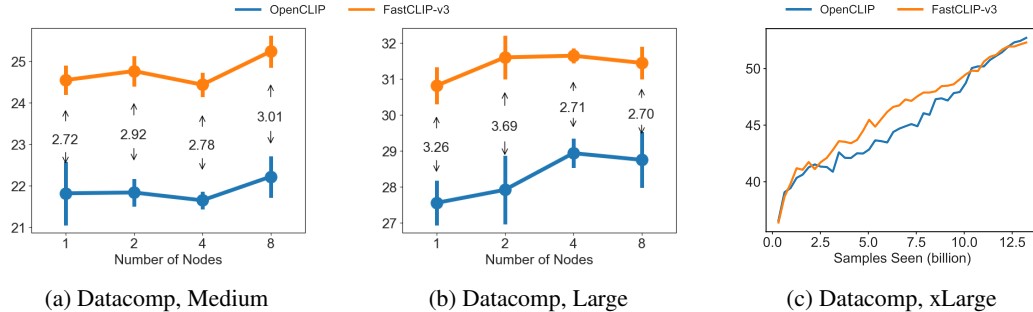

(a) Datacomp, Medium      (b) Datacomp, Large      (c) Datacomp, xLarge

Figure 10: Datacomp Avearge performance of OpenCLIP and FastCLIP-v3 in different settings. Subfigures (a), (b) present the results in the medium-scale and large-scale setting, with numbers denoting the improvement of FastCLIP-v3 over OpenCLIP. Subfigure (c) present the results in the xlarge-scale setting.

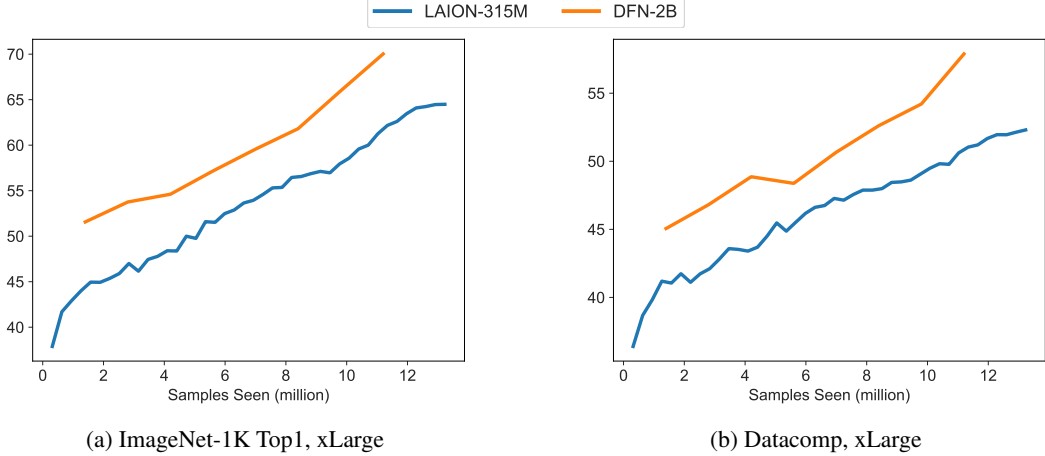

(a) ImageNet-1K Top1, xLarge      (b) Datacomp, xLarge

Figure 11: ImageNet-1k Top 1 accuracy (left) and Datacomp Average performance (right) of FastCLIP-v3 on different datasets in the xlarge-scale setting.

Table 15: Comparison between OpenCLIP and FastCLIP-v3 in terms of training time in the medium-scale setting. The shaded results are from FastCLIP-v3, and the others are from OpenCLIP. Computation denotes the whole computation time. Communication denotes the whole communication time. Pure Comm. denotes the communication time that is not overlapped with computation. Overlap denotes the overlapped time between computation and communication.

| Category | 1 Node | 2 Nodes | 4 Nodes | 8 Nodes |
|---|---|---|---|---|
| Total | 867.85 (11.04) | 880.19 (53.45) | 925.47 (27.77) | 1049.90 (32.44) |
|  | 866.36 (5.89) | 879.91 (52.17) | 917.54 (25.46) | 1028.06 (32.26) |
| Computation | 770.57 (6.10) | 738.87 (21.58) | 726.07 (1.53) | 742.93 (15.91) |
|  | 771.80 (5.53) | 737.93 (21.73) | 725.40 (2.01) | 742.90 (15.90) |
| Communication | 222.01 (4.43) | 403.40 (130.80) | 548.07 (60.97) | 698.87 (26.24) |
|  | 223.34 (5.51) | 400.76 (125.78) | 536.15 (59.29) | 675.43 (25.97) |
| Pure Comm. | 27.18 (1.61) | 68.74 (25.45) | 127.39 (30.29) | 224.71 (16.05) |
|  | 25.50 (2.24) | 64.32 (22.47) | 116.21 (28.48) | 200.97 (15.58) |
| Overlap | 194.84 (2.88) | 334.66 (105.36) | 420.68 (30.80) | 474.16 (10.23) |
|  | 197.84 (3.65) | 336.44 (103.35) | 419.94 (30.83) | 474.46 (10.41) |
| Others | 70.09 (8.17) | 72.58 (6.59) | 72.01 (2.73) | 82.26 (0.93) |
|  | 69.06 (1.67) | 77.66 (8.14) | 75.93 (2.83) | 84.19 (0.86) |

Table 16: Comparison between OpenCLIP and FastCLIP-v3 in terms of training time in the large-scale setting. The shaded results are from FastCLIP-v3, and the others are from OpenCLIP. The meaning of each category is the same as Table 15.

| Category | 1 Node | 2 Nodes | 4 Nodes | 8 Nodes |
|---|---|---|---|---|
| Total | 1125.29 (14.14) | 1234.06 (151.37) | 1396.76 (47.86) | 1564.46 (47.92) |
| | 1128.75 (9.75) | 1234.82 (153.86) | 1394.91 (48.35) | 1542.32 (47.87) |
| Computation | 960.14 (12.00) | 910.77 (10.48) | 891.71 (6.09) | 896.54 (8.02) |
| | 964.16 (9.10) | 910.94 (11.55) | 892.72 (4.72) | 897.59 (9.09) |
| Communication | 360.34 (15.55) | 655.30 (175.45) | 876.13 (71.52) | 1061.52 (55.08) |
| | 363.38 (16.66) | 652.78 (173.41) | 870.01 (69.56) | 1035.03 (56.84) |
| Pure Comm. | 56.73 (4.09) | 192.89 (129.45) | 379.10 (58.13) | 525.78 (57.22) |
| | 55.44 (2.23) | 190.56 (127.48) | 371.30 (55.62) | 498.95 (59.72) |
| Overlap | 303.62 (14.70) | 462.41 (46.02) | 497.02 (13.45) | 535.74 (2.33) |
| | 307.94 (18.14) | 462.22 (45.93) | 498.71 (13.97) | 536.08 (2.99) |
| Others | 108.42 (5.54) | 130.40 (12.26) | 125.95 (5.57) | 142.14 (2.08) |
| | 109.14 (2.67) | 133.33 (15.30) | 130.89 (4.34) | 145.78 (3.13) |

**Training Time of OpenCLIP and FastCLIP in Different Network Environments**: The results above (and in Section 6) are obtained from a cluster with InfiniBand interconnect. We conduct additional experiments on two different clusters with Slingshot interconnect. The results are presented below. It can be seen that our gradient reduction strategy has consistent improvement over the strategy used in OpenCLIP in different network environments.

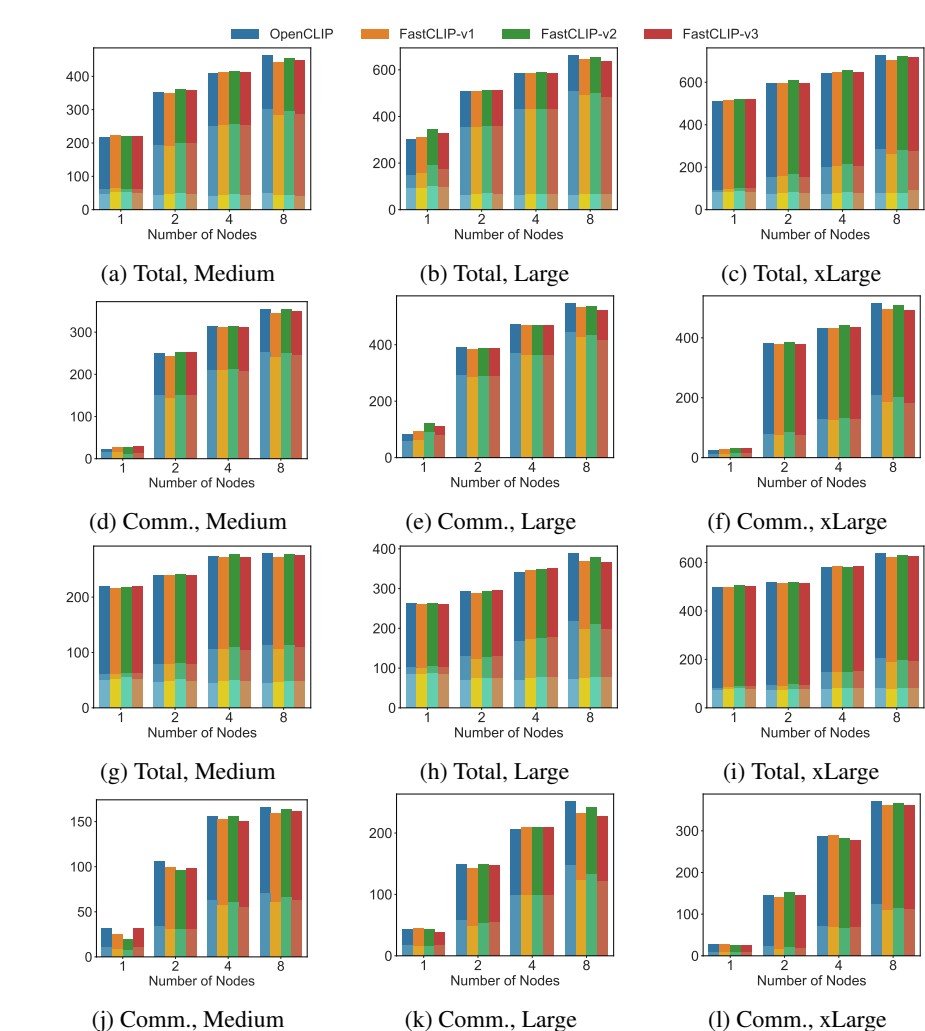

Figure 12: Plots of training time of different methods on two clusters with Slingshot interconnect. Subfigures (a) to (f) are results on one cluster, while Subfigures (g) to (l) are results on the other cluster. The meaning of each bar is the same as Figure 3.

The following tables present results used to plot the above figure.

Table 17: OpenCLIP vs. FastCLIP-v3 (shaded) in the medium-scale setting on Cluster 1 with Slingshot interconnect. The meaning of each category is the same as Table 15.

| Category | 1 Node | 2 Nodes | 4 Nodes | 8 Nodes |
|---|---|---|---|---|
| Total | 218.42 (14.87) | 352.99 (2.91) | 410.14 (0.94) | 462.18 (8.12) |
| | 221.46 (7.96) | 359.04 (12.21) | 412.67 (1.08) | 447.42 (4.18) |
| Computation | 157.03 (0.26) | 160.53 (0.09) | 160.11 (0.04) | 160.02 (0.09) |
| | 158.99 (1.75) | 160.88 (0.23) | 160.34 (0.06) | 160.35 (0.08) |
| Communication | 23.58 (13.49) | 250.21 (2.74) | 313.26 (2.33) | 354.59 (2.33) |
| | 29.88 (14.20) | 251.58 (12.11) | 312.32 (1.28) | 349.45 (4.46) |
| Pure Comm. | 14.28 (11.86) | 149.42 (2.71) | 210.57 (1.51) | 251.45 (2.11) |
| | 12.13 (5.20) | 150.19 (12.06) | 208.20 (1.29) | 245.39 (4.49) |
| Overlap | 9.31 (1.63) | 100.79 (0.03) | 102.69 (0.87) | 103.14 (0.25) |
| | 17.75 (9.04) | 101.39 (0.04) | 104.11 (0.09) | 104.06 (0.03) |
| Others | 47.12 (2.77) | 43.04 (0.27) | 39.45 (0.69) | 50.72 (9.91) |
| | 50.35 (1.03) | 47.97 (0.52) | 44.13 (0.38) | 41.69 (0.40) |

Table 18: OpenCLIP vs. FastCLIP-v3 (shaded) in the large-scale setting on Cluster 1 with Slingshot interconnect. The meaning of each category is the same as Table 15.

| Category | 1 Node | 2 Nodes | 4 Nodes | 8 Nodes |
|---|---|---|---|---|
| Total | 301.54 (22.83) | 510.42 (6.59) | 587.59 (7.53) | 660.99 (23.63) |
| | 329.86 (40.84) | 511.80 (6.53) | 586.17 (8.97) | 637.16 (13.73) |
| Computation | 153.12 (2.47) | 155.06 (0.05) | 154.63 (0.08) | 154.45 (0.05) |
| | 154.22 (3.44) | 155.53 (0.14) | 155.10 (0.05) | 155.19 (0.11) |
| Communication | 84.86 (23.74) | 389.90 (6.59) | 472.84 (6.84) | 545.75 (18.81) |
| | 110.76 (36.62) | 389.36 (6.43) | 467.79 (9.38) | 520.98 (15.02) |
| Pure Comm. | 58.00 (22.62) | 291.70 (6.63) | 371.31 (7.25) | 443.20 (18.99) |
| | 79.80 (35.59) | 288.68 (5.59) | 363.71 (8.58) | 416.43 (15.11) |
| Overlap | 26.86 (12.35) | 98.20 (0.07) | 101.53 (0.57) | 102.55 (0.87) |
| | 30.96 (17.49) | 100.68 (1.06) | 104.08 (0.88) | 104.55 (0.10) |
| Others | 90.42 (4.26) | 63.66 (1.09) | 61.66 (0.91) | 63.34 (4.74) |
| | 95.84 (7.48) | 67.59 (1.28) | 67.36 (0.49) | 65.54 (1.62) |

Table 19: OpenCLIP vs. FastCLIP-v3 (shaded) in the xlarge-scale setting on Cluster 1 with Slingshot interconnect. The meaning of each category is the same as Table 15.

| Category | 1 Node | 2 Nodes | 4 Nodes | 8 Nodes |
|---|---|---|---|---|
| Total | 511.28 (8.46) | 597.15 (3.50) | 643.54 (4.69) | 725.58 (35.32) |
| | 520.66 (6.96) | 597.52 (8.42) | 648.67 (6.48) | 717.43 (24.75) |
| Computation | 418.29 (0.59) | 442.58 (0.41) | 442.63 (0.19) | 441.86 (0.09) |
| | 419.27 (1.48) | 442.86 (0.10) | 442.99 (0.28) | 442.86 (0.17) |
| Communication | 24.52 (9.71) | 380.79 (2.92) | 432.24 (4.58) | 514.46 (32.87) |
| | 33.34 (12.27) | 378.48 (7.59) | 436.70 (6.90) | 492.55 (15.62) |
| Pure Comm. | 12.29 (7.00) | 79.79 (3.19) | 127.27 (4.15) | 207.18 (32.94) |
| | 16.68 (4.62) | 75.38 (6.89) | 127.03 (6.40) | 182.89 (15.91) |
| Overlap | 12.23 (2.73) | 301.00 (0.32) | 304.98 (0.65) | 307.28 (0.52) |
| | 16.66 (8.19) | 303.11 (0.94) | 309.67 (0.80) | 309.65 (0.69) |
| Others | 80.70 (1.53) | 74.78 (0.87) | 73.65 (0.59) | 76.54 (2.76) |
| | 84.71 (1.27) | 79.29 (1.84) | 78.65 (0.52) | 91.68 (13.55) |

Table 20: OpenCLIP vs. FastCLIP-v3 (shaded) in the medium-scale setting on Cluster 2 with Slingshot interconnect. The meaning of each category is the same as Table 15.

| Category | 1 Node | 2 Nodes | 4 Nodes | 8 Nodes |
|---|---|---|---|---|
| Total | 218.62 (1.49) | 239.22 (0.18) | 273.94 (1.10) | 278.21 (2.74) |
| | 219.67 (4.87) | 239.04 (2.73) | 271.29 (1.17) | 274.94 (4.29) |
| Computation | 157.47 (1.03) | 160.47 (0.14) | 167.51 (0.89) | 164.75 (0.05) |
| | 157.94 (1.08) | 160.85 (0.15) | 167.94 (1.33) | 164.89 (0.37) |
| Communication | 31.41 (6.16) | 105.83 (2.74) | 155.37 (1.14) | 165.85 (4.20) |
| | 31.12 (7.51) | 98.51 (2.14) | 150.03 (1.52) | 160.95 (3.00) |
| Pure Comm. | 10.63 (2.02) | 33.37 (1.17) | 62.29 (1.24) | 69.99 (2.41) |
| | 10.02 (2.23) | 30.41 (2.10) | 55.13 (2.08) | 62.52 (4.06) |
| Overlap | 20.78 (5.31) | 72.46 (1.63) | 93.08 (0.12) | 95.86 (1.96) |
| | 21.10 (5.28) | 68.10 (0.22) | 94.91 (0.91) | 98.43 (1.17) |
| Others | 50.52 (1.01) | 45.37 (0.88) | 44.14 (0.42) | 43.47 (0.88) |
| | 51.72 (2.28) | 47.78 (1.46) | 48.22 (0.60) | 47.53 (0.39) |

Table 21: OpenCLIP vs. FastCLIP-v3 (shaded) in the large-scale setting on Cluster 2 with Slingshot interconnect. The meaning of each category is the same as Table 15.

| Category | 1 Node | 2 Nodes | 4 Nodes | 8 Nodes |
|---|---|---|---|---|
| Total | 261.73 (5.81) | 293.85 (7.66) | 341.99 (6.88) | 387.59 (29.84) |
| | 260.15 (2.71) | 295.25 (5.15) | 350.30 (12.59) | 365.01 (31.42) |
| Computation | 158.96 (0.59) | 165.37 (1.34) | 174.27 (1.08) | 169.39 (0.47) |
| | 158.86 (0.71) | 166.24 (0.48) | 173.65 (2.33) | 168.31 (0.63) |
| Communication | 42.37 (6.16) | 149.25 (8.76) | 205.88 (3.59) | 250.92 (31.40) |
| | 38.83 (2.64) | 148.16 (3.75) | 208.68 (9.40) | 227.15 (31.55) |
| Pure Comm. | 17.33 (3.89) | 58.19 (5.73) | 98.24 (5.08) | 146.62 (32.25) |
| | 16.45 (0.70) | 54.01 (3.58) | 98.83 (11.70) | 120.91 (30.43) |
| Overlap | 25.04 (3.56) | 91.05 (3.12) | 107.63 (1.83) | 104.30 (1.00) |
| | 22.39 (2.91) | 94.14 (0.28) | 109.84 (2.31) | 106.24 (1.56) |
| Others | 85.44 (2.76) | 70.29 (0.78) | 69.48 (2.97) | 71.58 (3.20) |
| | 84.84 (2.27) | 74.99 (2.13) | 77.82 (3.60) | 75.79 (0.83) |

Table 22: OpenCLIP vs. FastCLIP-v3 (shaded) in the xlarge-scale setting on Cluster 2 with Slingshot interconnect. The meaning of each category is the same as Table 15.

| Category | 1 Node | 2 Nodes | 4 Nodes | 8 Nodes |
|---|---|---|---|---|
| Total | 496.14 (0.84) | 516.82 (5.65) | 581.00 (3.16) | 636.00 (16.93) |
| | 502.32 (5.29) | 515.99 (0.56) | 582.50 (2.89) | 626.40 (7.16) |
| Computation | 415.30 (0.10) | 422.40 (0.25) | 433.71 (1.03) | 433.66 (0.41) |
| | 415.34 (0.13) | 422.89 (0.21) | 432.92 (0.22) | 434.92 (0.48) |
| Communication | 26.49 (0.97) | 145.77 (7.14) | 287.66 (7.36) | 369.86 (12.53) |
| | 25.13 (2.47) | 144.39 (2.93) | 277.56 (1.98) | 362.24 (11.25) |
| Pure Comm. | 9.31 (0.46) | 23.58 (4.66) | 70.29 (1.96) | 123.79 (15.56) |
| | 8.65 (1.86) | 17.89 (0.62) | 67.45 (3.26) | 111.46 (6.13) |
| Overlap | 17.18 (0.61) | 122.19 (2.57) | 217.36 (8.75) | 246.07 (3.60) |
| | 16.48 (0.80) | 126.50 (2.31) | 210.11 (3.54) | 250.77 (5.46) |
| Others | 71.53 (0.55) | 70.84 (0.88) | 76.99 (2.15) | 78.56 (1.92) |
| | 78.33 (3.30) | 75.20 (0.55) | 82.14 (2.26) | 80.01 (0.87) |

