# OpenReview forum: "FastCLIP: A Suite of Optimization Techniques to Accelerate CLIP Training with Limited Resources"
_ICLR.cc/2025/Conference — Submitted to ICLR 2025_

### Official Review · Reviewer_rD79 · 2024-10-29

**Soundness:** 2
**Presentation:** 3
**Contribution:** 2
**Rating:** 3
**Confidence:** 4

**Summary:**

This paper introduced FastCLIP, which uses data-parallel for CLIP training. It claimed to (1) present an efficient gradient reduction strategy to reduce communication overhead, (2) compare inner LR schedule between constant and cosine schedule, (3) compare temperature param update methods and (4) compare optimizers.

**Strengths:**

1. This paper compared CLIP training performances with different temperature parameter updating methods and different optimizers.
2. The appendix provides detailed experiment results and parameter settings.
3. An introduced cosine schedule could provide better training accuracy.

**Weaknesses:**

1. Figure 3 is not so clear about the efficiency improvement. The difference between OpenCLIP and FastCLIP is not significant.
2. Figure 4 (a) does not show convergence on the accuracy curve which could be problematic.
3. Figure 4 (b) (c) do not show a significant difference in scalability.
4. The writing style needs improvement, and some part of this paper needs to be summarized. Also, the experiment part needs more details, the authors use many numbers in Table 3, 4, 5 but there is no information on what do they represent, accuracy, or some other metrics.
5. Lack of novelty. This paper compared multiple combinations of existing CLIP training techniques and proposed using data parallelization to scale up the training process. From this summary, existing techniques do not contribute much to novelty except cosine LR schedule, also, data parallelism mentioned in this paper does not show promising results compared to OpenCLIP.

**Questions:**

1. See weaknesses.
2. Is there any other significant difference between OpenCLIP and FastCLIP? Seems like they are also using data-parallel.
3. What do the number differences represent in Table 3, 4, 5, and Figure 2? The current comparison is not very straightforward.
4. Should the training accuracy converge?

---

> ### Author Response · Authors · 2024-11-23
>
> We sincerely thank the reviewer for their valuable review.
>
> **Q1**: Data parallelism mentioned in this paper does not show promising results compared to OpenCLIP (in Figure 3). Is there any other significant difference between OpenCLIP and FastCLIP?
>
> *A1*: We would like to note that we did not propose a new parallelism paradigm in this paper. Both OpenCLIP and FastCLIP leverages data parallelism. Instead, we proposed a novel gradient reduction strategy that eliminates part of the gradient communication. At 8 nodes, FastCLIP has a 1% to 3% reduction in terms of training time compared with OpenCLIP (please refer to Tables 15 to 22 in Appendix E, which present the per-iteration training time of FastCLIP and OpenCLIP on different scales and in different environments). Besides the gradient reduction strategy, another key difference between OpenCLIP and FastCLIP lies in the loss function, where OpenCLIP minimizes the Mini-Batch Contrastive Loss (MBCL) while FastCLIP optimizes the Global Contrastive Loss (GCL) family. Compare to OpenCLIP with MBCL, FastCLIP with GCL does not require a large batch size to obtain good performance, thus enabling researchers to train performant CLIP models in limited-resource settings.
>
> **Q2**: Figure 4 (a) does not show convergence on the accuracy curve which could be problematic. Should the training accuracy converge?
>
> *A2*: To our knowledge, large-scale ViT-based CLIP models requires a large amount of samples seen before convergence, which is not affordable even for researchers with a large number of resources. For example, in the official repository of Cherti el al. (2023): https://github.com/LAION-AI/scaling-laws-openclip/blob/master/figures.ipynb, from the last figure we can see that there are not clear signs of convergence after 34B samples seen. Thus in the literature, the common practice is to train the model for a given number of samples seen (e.g., 12.8B) to obtain a decent model. Curves similar to those in Figure 4 (a) in our paper can also be found in other works, e.g. Figure 1 in [1] and the figures in [2].
>
> [1] Hu et. al. Demystifying CLIP Data. ICLR 2024. https://openreview.net/forum?id=5BCFlnfE1g
>
> [2] open_clip. https://github.com/mlfoundations/open_clip/blob/main/docs/PRETRAINED.md
>
> **Q3**: Figure 4 (b, c) do not show a significant difference in scalability.
>
> *A3*: We would like to note that FastCLIP and OpenCLIP shares similar trend in Figure 4 (b, c). Moreover, FastCLIP only requires few resources (GPUs) to obtain good performance while OpenCLIP requires a large amount of them. As the curve in Figure 4 (b, c) predicts, the larger the number of nodes is, the worse the efficiency will be. This indicates that FastCLIP is at an advantage over OpenCLIP in terms of efficiency.
>
> **Q4**: The experiment part needs more details. What do the numbers represent in Table 3, 4, 5, and Figure 2?
>
> *A4*: We provided a brief introduction to the metrics we use in Section 5 (Line 347). To be specific, IN & Variants denotes the average top 1 accuracy on ImageNet-1K and 6 ImageNet distribution shift datasets; Retrieval denotes the average of mean recall at 1 on Flickr30K, MSCOCO and jaccard score on WinoGAViL; and Datacomp denotes the average performance on 38 tasks, which is a combination of accuracy on classification tasks and recall/ jaccard score on the three Retrieval tasks, as documented in Gadre et al. (2023). The numbers in the parentheses denote the standard deviation over 3 runs. Improvement denotes the absolute performance difference between FastCLIP-v3 and OpenCLIP on different metrics. We have revised our manuscript to include more detail about the metrics at Line 347 and at the captions of the tables and figure.
>
> **Q5**: FastCLIP is built upon existing techniques and is not novel.
>
> *A5*: We refer the reviewer to Q1 in the general response for explanation.
>
> **Q6**: The writing style needs improvement, and some part of this paper needs to be summarized.
>
> *A6*: We refer the reviewer to Q2 in the general response for explanation.

---

### Official Review · Reviewer_rov8 · 2024-10-31

**Soundness:** 3
**Presentation:** 3
**Contribution:** 2
**Rating:** 5
**Confidence:** 3

**Summary:**

This paper introduces a distributed framework designed to improve the efficiency of CLIP by optimizing resource use and reducing dependency on large batch sizes and numerous GPUs. They use several optimization techniques, including a gradient reduction strategy and a flexible LR schedule, to facilitate efficient CLIP training with limited resources. The study compares FastCLIP to OpenCLIP and demonstrates consistent performance improvements on different scales of data and compute resources.

**Strengths:**

- Efficient resource utilization for settings with limited computational resources, making CLIP training more accessible.

- Systematic Optimizations: The paper provides a structured approach to optimizing multiple aspects of CLIP training, including LR schedules, temperature parameter updates, and gradient communication strategies.

- Experimental Validation: Comprehensive testing on various datasets and compute scales effectively shows the benefits of FastCLIP compared to OpenCLIP.

**Weaknesses:**

- Limited Novelty in Optimization Techniques: The optimization techniques applied, such as learning rate decay and gradient reduction, are well-established and may not contribute novel methodological insights.

- Resource Comparison Limitations: The paper does not conduct an extensive ablation study for larger datasets due to resource constraints, which may limit the generalizability of results on extremely large data scales.

- Assumption of Availability of Multiple GPUs: Although designed for resource-limited environments, FastCLIP still requires access to multiple GPUs, which may limit applicability for highly resource-constrained users.

**Questions:**

How does the performance of FastCLIP vary across different types of GPUs or compute environments?

Could other advanced optimization techniques (e.g., adaptive optimizers beyond AdamW) further enhance the framework's efficiency?

Would additional experiments on extremely large datasets (>1 billion samples) align with the current findings for smaller datasets?

---

> ### Author Response · Authors · 2024-11-23
>
> We sincerely thank the reviewer for their valuable review.
>
> **Q1**: The authors did not conduct extensive ablation study on large scale datasets due to resource constraints, which may limit the generalizability of results on extremely large data scales. Would additional experiments on extremely large datasets (>1 billion examples) align with the current findings for smaller datasets?
>
> *A1*: We are conducting an experiment of FastCLIP-v3 on a 1.4B subset of the DFN-2B dataset (Fang et al. 2023). The number of samples finished at this moment is 11.2B. We compare the performance of FastCLIP-v3 on this data to that trained on LAION-315M reported in the paper in the following table, where the ImageNet-1K top1 accuracy at different number of samples seen are presented. We can see that our approach is still effective on extremely large-scale data. We also plot the performance curves of FastCLIP-v3 on different datasets and present it in Appendix E.2 of the revision. We expect to finish the training with 12.8B samples seen and add the result in the next revision.
>
> | Work | Data | Acc. (2.8B) | 5.6B | 8.4B | 11.2B |
> | --- | --- | --- | --- | --- | --- |
> | Ours | DFN-1.4B | 53.76 | 57.16 | 61.81 | 70.3 |
> | Ours | LAION-315M | 47.01 | 51.52 | 56.56 | 60.1 |
>
>
> **Q2**: FastCLIP still requires access multiple GPUs, which may limit applicability for highly resource-constrained users.
>
> *A2*: We would like to note that compared with OpenCLIP which requires tens of to hundreds of GPUs, FastCLIP already makes a big step at reducing the number of GPUs for large-scale CLIP training (8 in our experiments). In our paper, we experimented with hundred millions of data. Hence, in order to finish our extensive experiments we use 8 to 16 GPUs. We believe this level resources can be easily available to many researchers in the community. It is an interesting direction to further lower down the resource requirement to accommodate the need of highly resource-constrained users.
>
> **Q3**: How does the performance of FastCLIP vary across different types of GPUs or compute environments?
>
> *A3*: In terms of downstream performance (e.g. zero-shot classification and retrieval), the results are not dependent on types of GPUs or compute environments. In terms of training time, we profile the results on different clusters with different types of GPUs and interconnect, which are presented in Section 6 and Appendix E.2. FastCLIP-v3 demonstrates consistent improvement over OpenCLIP under different settings.
>
> **Q4**: Could other advanced optimization techniques (e.g., adaptive optimizers beyond AdamW) further enhance the framework's efficiency?
>
> *A4*: In this work we tested 4 optimizers on different scales and found that AdamW achieves the best overall performance. There is possibility that other optimiers outperform AdamW for CLIP training. It is an important research question and we leave it for future work.
>
> **Q5**: The gradient reduction strategy and optimization techniques lack novelty.
>
> *A5*: We refer the reviewer to Q1 in the general response for explanation.

---

### Official Review · Reviewer_5GWx · 2024-11-04

**Soundness:** 2
**Presentation:** 2
**Contribution:** 2
**Rating:** 3
**Confidence:** 3

**Summary:**

This paper introduces FastCLIP, a distributed training framework designed to optimize CLIP model training using compositional optimization techniques, removing the dependency on large batch sizes for effective model performance. The framework incorporates an efficient gradient reduction strategy to minimize communication overhead and conducts comprehensive ablation studies on various components, including learning rate schedules (constant vs. cosine), temperature parameter update rules, and different optimizers (AdamW, LAMB, Lion, and SGDM).

**Strengths:**

1. This paper studies an important problem in CLIP training, "how to efficiently and effectively training CLIP models with limited resources".

2. This paper designs an efficient distributed training framework based on advanced compositional optimization techniques. It conducts ablation studies on the several key components during training, such as the update rule of learning rate, temperature parameter and model papers, providing valuable insights for future work in optimizing large-scale model training.

3. Experimental results across various compute and data scales demonstrate that FastCLIP significantly outperforms existing methods, such as OpenCLIP, enhancing training efficiency on setups with up to 32 GPUs.

**Weaknesses:**

1. This paper is not designed for general CLIP training but is instead built on the assumptions and techniques from prior work[1]. As a result, its applicability may be limited to scenarios that align with these specific assumptions, restricting its generalizability to broader CLIP training tasks.

2. The paper primarily builds on existing techniques, such as compositional optimization and gradient reduction strategies, without introducing fundamentally new concepts. While it refines and optimizes these methods for CLIP training, the core ideas themselves are not particularly novel.

3. The paper's writing lacks clarity, making it difficult to follow at times. The structure of the paper would benefit from significant revision to improve its readability and logical consistency.

4. The experiment compares FastCLIP with OpenCLIP, which is designed for large-scale clusters and requires large batch sizes. In contrast, FastCLIP is tested on a relatively smaller cluster (up to 32 GPUs), violating the original conditions under which OpenCLIP was evaluated. Additionally, the results in experiments 4 and 5 are inconsistent, with different methods showing unstable performance. This lack of stability weakens the conclusions and limits the insights that can be drawn for future work.

[1] Yuan Z, Wu Y, Qiu Z H, et al. Provable stochastic optimization for global contrastive learning: Small batch does not harm performance. ICML 2022.

**Questions:**

See the weaknesses.

---

> ### Author Response · Authors · 2024-11-23
>
> We sincerely thank the reviewer for their valuable review.
>
> **Q1**: The applicability of FastCLIP is limited due to the assumptions made in [1].
>
> *A1*: We respectfully disagree with the reviewer. [1] has demonstrated its effectiveness of using a small batch size compared with previous approaches, e.g., SimCLR. This optimization technique has been demonstrated to be effective across a broad range of problems (e.g., LibAUC [2]). The assumptions made in [1] are fairly standard for analyzing the convergence of optimization algorithms. We noted that most existing papers of CLIP training may focus on other perspectives but simply use existing optimization algorithms e.g., Adam. However, the convergence analysis for any optimization algorithms would require some assumptions of the problem, and this does not prevent them from being applied in practice. Similarly, the effectiveness of our framework FastCLIP has been demonstrated compared with OpenCLIP with a small batch size through extensive experiments across various compute and data scales.
>
> [1] Yuan Z, Wu Y, Qiu Z H, et al. Provable stochastic optimization for global contrastive learning: Small batch does not harm performance. ICML 2022.
>
> [2] Yuan Z, Zhu D, Qiu Z H, et al. LibAUC: A Deep Learning Library for X-risk Optimization. KDD 2023.
>
> **Q2**: OpenCLIP is designed for large-scale clusters and large batch sizes, but the authors compared FastCLIP and OpenCLIP in only small batch settings, violating the original conditions under which OpenCLIP was evaluated.
>
> *A2*: OpenCLIP is the state-of-the-art distributed training framework for CLIP training. We addressed a different scientific question "how to train a CLIP model on billion-scale data with limited compute resources?" Although this is not a main-stream approach as industry leads the research of CLIP training with large compute resource, we believe it is a fundamental problem that requires rigorous research. To the best of our knowledge, there are no existing distributed training algorithms of CLIP training that work well with a small batch size. Hence, our work would pave the way for latter works on accelerating CLIP training with limited compute resources. We would like to point out  that results in Table 6 showed that FastCLIP with small batch size already has comparable performance to OpenCLIP with large batch size.
>
> **Q3**: The results in Table 4 and 5 are inconsistent, with different methods showing unstable performance.
>
> *A3*: We would like to note that Table 4 shows the results of FastCLIP-v0 to -v3 with the AdamW optimizer, while Table 5 shows the results of FastCLIP-v3 with different optimizers. Thus the 4th and 8th rows of both tables overlap with each other, which are identical and consistent. In Table 4, FastCLIP-v3 achieves 1st place in 4 of the 6 metrics (3 in the medium-scale setting and 3 in the large-scale setting), and 2nd place in the remaining 2 metrics. Thus we arrive at the conclusion that FastCLIP-v3 is the overall most-performant method for CLIP training in the Global Contrastive Learning framework. The same reasoning applies to the AdamW optimizer in Table 5.
>
> **Q4**: FastCLIP is built upon existing techniques and is not novel.
>
> *A4*: We refer the reviewer to Q1 in the general response for explanation.
>
> **Q5**: The writing of this paper lacks clarity.
>
> *A5*: We refer the reviewer to Q2 in the general response for explanation.

---

### Author Response · Authors · 2024-11-23
**General Response to Common Questions Raise by Reviewers**

We sincerely thank all reviewers for their valuable review. Here we address common questions raised by the reviewers, then we address each reviewer's questions in their individual rebuttal.

**Q1**: Novelty of the FastCLIP framework.

*A1*: The key novelty of FastCLIP lies at the **distributed infrastructure** to support algorithms that optimize **Global Contrastive Loss** of contrastive learning. FastCLIP is different from existing open-soured distributed frameworks for CLIP training, e.g., OpenCLIP. In particular, OpenCLIP supports algorithms based on mini-batch contrastive losses. However, these algorithms require a large batch size to work well. Recent advances have proposed efficient algorithms for optimizing global contrastive losses, including SogCLR and iSogCLR. But there is lack of distributed infrastructure to support them for training on billions of data points.  In this distributed infrastructure,
- we proposed a novel gradient reduction strategy, which reduces the communication costs of OpenCLIP.
- we designed better optimization strategies for different components within the framework.

**Q2**: Writing and exposition of the paper.

*A2*: This paper is presented in the following manner. First we introduce the overall design and the gradient reduction strategy of the FastCLIP framework. Next, we conduct extensive experiments comparing new strategies for different components within the framework with existing strategies. In the end, we demonstrate the scaling performance of our most performant algorithm FastCLIP-v3 and compare it with OpenCLIP at different scales to show the effectiveness of FastCLIP in limited-resource settings. We added a paragraph in the revision describing the exposition at the end of Introduction to help readers follow the paper.

**Q3**: Results of FastCLIP-v3 on extremely large datasets (>1 billion examples).

*A3*: We are conducting an experiment of FastCLIP-v3 on a 1.4B subset of the DFN-2B dataset (Fang et al. 2023). The number of samples we finished at this moment is 11.2B. We compare the performance of FastCLIP-v3 on this data to that trained on LAION-315M reported in the paper in the following table, where the ImageNet-1K top1 accuracy at different number of samples seen are presented. We can see that our approach is still effective on extremely large-scale data. We also plot the performance curves of FastCLIP-v3 on different datasets and present it in Appendix E.2 of the revision. We expect to finish the training with 12.8B samples seen and add the result in the next revision.

| Work | Data | Acc. (2.8B) | 5.6B | 8.4B | 11.2B |
| --- | --- | --- | --- | --- | --- |
| Ours | DFN-1.4B | 53.76 | 57.16 | 61.81 | 70.3 |
| Ours | LAION-315M | 47.01 | 51.52 | 56.56 | 60.1 |

---

### Meta-Review · Area_Chair_vaAw · 2024-12-20

**Metareview:**

Thank you for your submission to ICLR. This paper introduces FastCLIP, a set of techniques and implementation for efficient training of CLIP in settings with limited resources. To carry this out, the authors developed a gradient reduction strategy to reduce communication requirements, and then investigated aspects of the optimization procedure (e.g., inner learning rate schedule, update rule of the temperature/model parameters).

The reviewers agree with the motivation for this paper, and all appreciate the thorough set of empirical results and ablations. However, the reviewers also had a number concerns, including:
- Limited novelty of the proposed techniques.
- In some cases, there was not a great deal of demonstrated improvement of the proposed method over existing baselines.
- Issues with clarity in the writing and in certain figures.

I encourage you to carefully consider and incorporate these comments from reviewers upon resubmission.

**Additional Comments On Reviewer Discussion:**

The authors disagreed with and argued against the criticisms listed above (e.g., limited novelty, lack of paper clarity), but the reviewers remained unconvinced. After rebuttal and discussion, the reviewers maintained their scores.

---

### Decision · Program_Chairs · 2025-01-22

Reject